# Compute Where it Counts: Self Optimizing Language Models

Yash Akhauri [1]  Mohamed S. Abdelfattah [1]

## Abstract

Efficient LLM inference research has largely focused on reducing the cost of each decoding step (e.g., using quantization, pruning, or sparse attention), typically applying a uniform computation budget to every generated token. In practice, token difficulty varies widely, so static compression can over-compute on easy steps and under-compute on hard ones. We study *dynamic budget allocation* for autoregressive decoding: learning how much computation to spend *per token* from within a single model.

Self-Optimizing Language Models (SOL) pair a frozen LLM with a lightweight policy network that reads the LLM hidden state and selects a discrete *efficiency action* at each decode step. Actions can jointly control (i) token-level attention sparsity, (ii) structured activation pruning in the MLP, and (iii) activation quantization bit-width, while leaving the base model weights unchanged. We train the policy with group-relative policy optimization on teacher-forced episodes: the token sequence is fixed, while we sample multiple compute schedules (i.e., "counterfactual" schedules that vary only the efficiency actions for the same token path) and compare their likelihoods under the same supervision. Our reward trades off language-model quality against soft penalties that encourage episode-average budget usage to match a requested target. Across model variants and compute regimes, SOL improves quality at matched budget over static allocation and strong random schedule search, offering a complementary axis for inference-efficiency optimization. SOL discovers a better quality-efficiency Pareto frontier across all our experiments and improves MMLU accuracy by up to 7.3% over uniform budget allocation strategies. [Code]

[1]Cornell University. Correspondence to: Yash Akhauri <ya255@cornell.edu>.

*Proceedings of the $43^{rd}$ International Conference on Machine Learning*, Seoul, South Korea. PMLR 306, 2026. Copyright 2026 by the author(s).

## 1. Introduction

Deploying LLMs at scale has driven efficiency research to make each decoding step cheaper via quantization (Chen et al., 2025; Hooper et al., 2024; Liu et al., 2024b; Zhao et al., 2024), pruning (Liu et al., 2023b; Akhauri et al., 2024; Child et al., 2019), low-rank compression (Chang et al., 2025a;b; Zhang et al., 2024), sparse attention patterns (Choromanski et al., 2020; Zaheer et al., 2020; Akhauri et al., 2025a; Zhang et al., 2023; Li et al., 2024), speculative decoding (Leviathan et al., 2023), and related techniques. Most of these methods, however, apply essentially the same compute budget to every generated token. In practice, token difficulty varies widely: some steps are locally predictable, while others depend on long-range context or precise intermediate computations. A fixed per-token budget therefore tends to *over-compute* on easy steps and *under-compute* on harder steps.

A fixed per-token budget often causes the model to either over-compute or under-compute. For example, with token sparsity, some tokens require longer-range dependencies whereas others do not, providing an opportunity to dynamically determine computation budget through the number of tokens attended to with each generated token. This also applies to other LLM compression methods such as quantization, weight sparsity, and low-rank factorization. Controlling the compression rate based on token generation difficulty has the potential to unleash a new axis of efficiency in LLMs.

One main challenge is that per-token efficiency decisions interact across time. Each generated token can be attended to by future tokens, therefore, aggressively compressing or dropping information that seems safe at step $t$ can cause degradation in quality several steps later. This motivates treating compute allocation as a sequential decision problem rather than a per-step or step-agnostic heuristic. To address this, we propose Self-Optimizing Language Models (SOL). The base LLM is kept frozen and we add a lightweight policy model that reads the LLM hidden state and simple progress/budget features, then selects a discrete efficiency action at each decoding step. Each action instantiates a compute regime inside the same model by controlling (i) token-level attention sparsity, (ii) structured MLP activation pruning, and/or (iii) activation quantization bit-width—without changing the base weights.

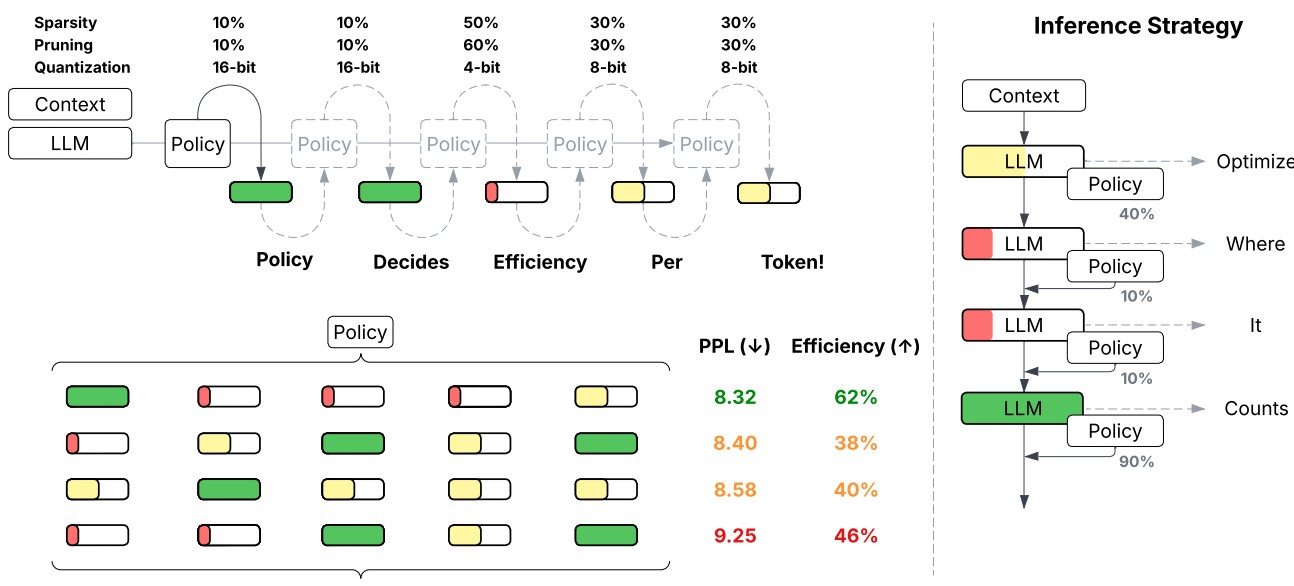

*Figure 1.* (**Top Left**) The policy is a small autoregressive transformer that uses the LLM's own hidden state to decide how much compute to allocate *per token*. (**Bottom Left**) At training time, the policy samples multiple counterfactual compute schedules (trajectories) and learns to optimize quality under a requested budget target. (**Right**) At inference, the controller incurs a minimal, constant overhead (proportional to the control horizon, e.g., 16 decode steps), enabling fine-grained control over per-step efficiency.

A key design goal is deployment-time controllability. Rather than hard-coding a single operating point, we condition the policy on a requested compute budget. During training, we sample per-sequence budget targets and provide them as part of the policy observation, so a single policy can learn to operate across a range of compute budgets. We train the policy using group-relative policy optimization (GRPO) with teacher-forced episodes: for each input we sample multiple counterfactual compute schedules that share the same token path, and optimize a reward that balances language-model quality against soft penalties that encourage the episode-average budget usage to match the requested targets. Our contributions are:

- We introduce *learnable per-token budget allocation* for LLM decoding: a lightweight policy allocates compute at each step while keeping the base LLM frozen.

- Our policy acts on a unified action space that controls multiple efficiency mechanisms: token-level attention sparsity, structured MLP activation pruning, and activation quantization. SOL tracks a better quality-efficiency Pareto frontier when compared to fixed budget allocation across all our experiments.

- We present a practical training recipe based on group-relative policy optimization with teacher-forced counterfactual trajectories, conditioning the policy on requested budget targets and using soft budget-matching

penalties to achieve controllable efficiency and model quality trade-offs.

## 2. Related Work and Motivation

Prior research on methods to optimize LLM inference focuses on **(i) reducing the memory and computational** footprint of the LLM, by quantizing the weights, activations (Lee et al., 2022; Huang et al., 2021; Dotzel et al., 2023), or pruning linear projections and feed-forward network (FFN) layers (Liu et al., 2023b; Akhauri et al., 2024; Feng et al., 2024), **(ii) context compression** by sparsifying the attention itself, using importance-based techniques to retain only important tokens in the KV-Cache (Zhang et al., 2023; Li et al., 2024; Xiao et al., 2023; Liu et al., 2023a; Xiao et al., 2024), or exploiting the low-rank nature of the KV-Cache to reduce its memory footprint (Chang et al., 2025a;b). **(iii) reducing memory bandwidth** by employing query-aware sparsity (Akhauri et al., 2025a; Tang et al., 2024; Wu et al., 2024). These methods typically apply a uniform budget to every token, meaning, the compression level is the same for every generated token.

A second line of work focuses on *how much* compute to spend per token, but skipping layers of computation. These "early-exit" methods dynamically decide how many layers to execute for each token, stopping computation once a confidence criterion is met; this predates transformers and has been applied to convolutional networks as well. In sequence-to-sequence and autoregressive LMs, this "depth-adaptive"

execution is orthogonal to our focus: we keep the depth fixed and instead adapt how much context, precision, and sparsity to use for each generated token. (Schuster et al., 2022; Elbayad et al., 2019; Elhoushi et al., 2024) introduce token-wise early exit strategies. (Sukhbaatar et al., 2019) demonstrates that later layers actually demonstrate a higher attention span than earlier layers, and over long-decode tasks, early-exit of several tokens can harm the global-attention pattern as tokens that exit early do not explicitly commit to KV-Cache of subsequent layers. Beyond early exit of tokens, recent research has also proven that splitting token generation across models (Akhauri et al., 2025b; Yu et al., 2025; Fu et al., 2025; Li & Goyal, 2025) by learning to offload difficult tokens or spans to larger models can significantly improve reasoning performance.

Our contribution is a controller that operates inside a single frozen LLM, allocating *per-token* inference compute by selecting discrete actions over multiple compression methods (token-level attention sparsity, structured MLP activation pruning, and activation quantization). Unlike early-exit / layer-skipping approaches that adapt *depth*, SOL modulates *how* each decode step is executed through these orthogonal efficiency knobs. We train the controller to reliably *hit a requested budget* even in large, combinatorial action spaces by conditioning on budget targets and optimizing over teacher-forced, counterfactual compute schedules—motivated by the fact that the effectiveness (and downstream KV impact) of sparsity/quantization/pruning is token- and context-dependent.

## 3. Background

**Decoding is a stateful process.** A transformer decoder maintains a *key–value (KV) cache* that accumulates per-layer representations for previously processed tokens. At step $t$, the model produces the next-token distribution from the current token and by attending to $\{(K_\ell^{(j)}, V_\ell^{(j)})\}_{j<t, \ell\in[1..L]}$ in the cache, where $L$ is the number of decoder layers. The choice of computation at step $t$ (full, sparse attention, pruned heads, low-precision arithmetic) does not only determine the next token produced, but also impacts the committed KV-Cache entry for that token, which future steps will attend to. If a step is decoded with *less information*, its committed KV states deviate from an identical, dense model. We refer to this deviation as **KV-pollution**, where the tokens' cached states are approximated, harming future predictions when those states become important for the current query. This phenomenon is not specific to attention sparsity; it also arises when we skip layers, prune heads (missing KV-Cache for that token on layers or heads) or prune/quantize neurons (lower quality of the hidden-state for that token). Further, this impact is *delayed* by nature: the next-token loss may appear unaffected because the model relied on its

entire past context, however, the effect of this approximation may appear several steps later when future tokens attend to the approximated token. Empirically, even short bursts of compression can cause delayed perplexity spikes after returning to dense decoding, with token sparsity exhibiting the strongest tail effects (Appendix A, Fig. 7).

**Episodes and control horizon.** The impact of an efficiency decision can materialize several steps later (e.g., through degraded KV entries that future tokens attend to), so purely myopic, single-step feedback at time $t$ can underestimate the true cost of aggressive compression. We therefore cast decoding-time control as short *episodes* of length $T$ (default $T=16$): after a dense prefill initializes a clean cache and state $s_0$, the policy selects a sequence of discrete actions $a_t \in \{1, \ldots, A\}$ for $t=1, \ldots, T$. Each action instantiates that efficiency method inside the frozen LLM via per-step knobs $(\kappa(a_t), \rho(a_t), \eta(a_t))$ (e.g., attention keep fraction, MLP keep fraction, and normalized precision), giving the next state $s_{t+1} = f(s_t, a_t)$ and an episode trajectory $\tau = (s_0, a_1, \ldots, a_T)$ whose objective aggregates reward over the horizon, e.g.,

$$R(\tau) = \sum_{t=1}^{T} r(s_t, a_t),$$

enabling credit assignment for delayed effects within the episode.

Continuing indefinitely under aggressive actions can compound KV-pollution, so we bound its persistence with *periodic KV-cache refresh*: after every $T$ decode steps, we optionally run a fast, fully-dense pass over the already-known last $T$ tokens to rebuild their KV states to their dense equivalents before starting the next episode; this refresh recomputes cache entries but does not resample or regenerate tokens. In practice, prefill is substantially higher-throughput than token-by-token decode, so this refresh has limited overhead. Unless mentioned otherwise, we use $T=16$ to align with page-based inference settings and to balance two goals: **(i)** limiting the lifetime of polluted cache entries, and **(ii)** providing a long enough horizon for the policy to learn non-myopic trade-offs; $T$ can be increased (or refresh removed) if optimizing for different deployment constraints such as memory footprint. This is consistent with our KV-pollution measurements showing delayed degradation even after switching back to dense compute (Appendix A, Fig. 7). The refresh period is a deployment knob rather than a requirement of SOL: our horizon study evaluates $T \in \{4, 16, 64\}$ and shows that the policy advantage over fixed allocation persists, and in fact becomes stronger, as the horizon increases.

# 4. Method

We use a language model with parameters $f_\theta$ and learn a lightweight policy $\pi_\phi$ that allocates inference time compute *per decode step*. The base model parameters $\theta$ are never updated; only the parameters of the policy $\phi$ are trained.

**Notation.** We optimize over *episodes* of $T$ decode steps (the *control horizon*). A *compute schedule* is an action sequence $a_{1:T}$, where each action selects a compute regime inside the frozen LLM. For each input/prefix, we sample $K$ counterfactual compute schedules under teacher forcing (the GRPO *group size*) and update the policy by comparing these schedules.

**Policy inputs and architecture.** We decode episodes of length $T$ (we use $T{=}16$). At each decode step $t \in \{1, \ldots, T\}$, the policy is fed: (i) the final-layer hidden state from the previous step $h_{t-1}$, i.e., the output of the last transformer block before the LM head for token $x_{t-1}$, and (ii) the embedding of the current input token $e(x_t)$. In addition, the policy receives an 8-dimensional scalar feature vector $s_t \in \mathbb{R}^8$ that makes the budget control explicit:

$$
s_t = \begin{bmatrix} \underbrace{t/T}_{\text{progress}}, & \underbrace{\mathbf{1}_{\text{eff},t}}_{\text{effective-step flag}}, & \underbrace{C_\kappa,\ C_\rho,\ C_\eta}_{\text{requested targets}} \\ \underbrace{\bar{\kappa}_{<t} - C_\kappa,\ \bar{\rho}_{<t} - C_\rho,\ \bar{\eta}_{<t} - C_\eta}_{\text{running deviations from target}} \end{bmatrix}.
$$

Here $\mathbf{1}_{\text{eff},t} \in \{0,1\}$ indicates whether step $t$ counts toward the budget. In our sparse attention implementation, we always keep $T_s$ *sink* tokens and the most recent $T_w$ *window* tokens dense (we use $T_s{=}4$ and $T_w{=}2$ in our main experiments; Appendix B); when the controllable region beyond these always-dense tokens is empty (e.g., for very short prefixes), the chosen action has no effect and we set $\mathbf{1}_{\text{eff},t} = 0$. In our main experiments with a 1024-token dense prefill, $\mathbf{1}_{\text{eff},t} = 1$ for all $t$. The quantities $\bar{\kappa}_{<t}, \bar{\rho}_{<t}, \bar{\eta}_{<t}$ are running averages over previous effective steps in the episode (defined below; if no effective step has occurred yet, we set these running averages to the corresponding targets).

The policy is a small autoregressive transformer that maintains a KV cache across steps within the episode; it also conditions on the previous action via a learned action embedding. At each step it outputs logits over the discrete action set. During training, we sample actions $a_t \sim \pi_\phi(\cdot \mid h_{t-1}, e(x_t), s_t)$ with a temperature of 1.3. Unless stated otherwise, we select actions greedily at evaluation time: $a_t = \arg\max_a \pi_\phi(a \mid h_{t-1}, e(x_t), s_t)$. The policy state (KV cache of the policy itself) is reset at the start of every episode. In our largest setting (Llama-3.1-8B-Instruct), the controller has 8.56M parameters, making it lightweight relative to the frozen base model.

**Actions and controlled compute.** Each discrete action $a \in \{1, \ldots, A\}$ corresponds to a tuple of compute knobs in the available action set $a \mapsto (\kappa(a), \rho(a), q(a))$, where:

- $\kappa(a) \in [0,1]$ is the token-attention keep fraction (token sparsity),

- $\rho(a) \in [0,1]$ is the structured activation keep fraction (channel pruning in the MLP),

- $q(a) \in \{5, \ldots, 16\}$ is the activation quantization bit-width (we use the normalized ratio $\eta(a) = q(a)/16 \in [0,1]$).

Unless stated otherwise, the same action tuple $(\kappa, \rho, q)$ is broadcast uniformly across all transformer layers at a given decode step. Thus, this paper studies *temporal* compute allocation across decode steps rather than per-layer or per-head allocation. A per-layer formulation is compatible with SOL, but would substantially enlarge the action space and predictor cost, so we leave it to future work.

The action space may include any subset of these axes; if an axis is not enabled (only one possible value), it simply becomes constant and incurs no budget penalty. We use "keep fraction" here for clarity and consistency across different compression methods. We train under teacher forcing: within each episode the token sequence is fixed and only the *efficiency actions* vary across counterfactual compute schedules. Let $p_t^{a_t}(\cdot)$ be the next-token distribution produced by the frozen LLM at step $t$ when executing action $a_t$, and let $y_t$ denote the ground-truth next token. We define the per-step task reward as the token log-likelihood:

$$
r_t^{\text{task}} = \log p_t^{a_t}(y_t) = -\text{CE}\big(p_t^{a_t}, y_t\big). \tag{1}
$$

To encourage the policy to meet a requested compute target, we penalize deviations of the *episode-average* realized compute from the requested budgets. Let $\mathbf{1}_{\text{eff},t} \in \{0,1\}$ indicate whether step $t$ counts toward the budget (i.e., whether the controllable region beyond the always-dense sink/window tokens is non-empty). For an episode of length $T$, define the realized averages

$$
\bar{\kappa} = \frac{\sum_{t=1}^T \mathbf{1}_{\text{eff},t}\, \kappa(a_t)}{\sum_{t=1}^T \mathbf{1}_{\text{eff},t}}, \quad \bar{\rho} = \frac{\sum_{t=1}^T \mathbf{1}_{\text{eff},t}\, \rho(a_t)}{\sum_{t=1}^T \mathbf{1}_{\text{eff},t}},
$$
$$
\bar{\eta} = \frac{\sum_{t=1}^T \mathbf{1}_{\text{eff},t}\, \eta(a_t)}{\sum_{t=1}^T \mathbf{1}_{\text{eff},t}}.
$$

where $\kappa(a_t)$ is the token-attention keep fraction, $\rho(a_t)$ is the structured activation keep fraction (e.g., MLP channel keep-rate), and $\eta(a_t)$ is the normalized activation precision (e.g., $\eta = q/16$ for $q$-bit quantization).

Let $(C_\kappa, C_\rho, C_\eta)$ be the requested targets (provided to the policy as inputs). We use a tolerance band $\tau$ and a squared

hinge penalty outside the band:

$$\psi(\Delta; \tau) \; = \; \big( \max\{0, |\Delta| - \tau\} \big)^2.$$

The episode-level compute penalty is

$$\mathcal{C} \; = \; \alpha_\kappa \, \psi(\bar\kappa - C_\kappa; \tau) + \alpha_\rho \, \psi(\bar\rho - C_\rho; \tau) + \alpha_\eta \, \psi(\bar\eta - C_\eta; \tau), \tag{2}$$

with nonnegative trade-off weights $(\alpha_\kappa, \alpha_\rho, \alpha_\eta)$. This objective encourages the policy to *match* the requested budgets (within tolerance), rather than simply minimizing compute.

Because efficiency decisions can have delayed effects within an episode, we use a discounted *return-to-go* over the task rewards to assign credit:

$$G_t \; = \; \sum_{u=t}^{T} \gamma^{u-t} \, r_u^{\text{task}}, \tag{3}$$

where $\gamma \in (0, 1]$ is a discount factor (we use $\gamma = 0.85$). We then combine task return and compute penalty into the per-step signal used for GRPO:

$$r_t \; = \; G_t \; - \; \mathcal{C}. \tag{4}$$

$\mathcal{C}$ is computed once per trajectory (compute schedule) from the episode-average gaps and broadcast across steps.

In all experiments, the token-sparsity knob $\kappa$ is instantiated with **Quest** (Tang et al., 2024) by selecting a budgeted subset of KV pages and applying an additive $-\infty$ mask to dropped keys. The structured pruning knob $\rho$ is implemented as **TEAL-style** (Liu et al., 2024a) activation pruning: per token, we keep the top-$\lceil \rho \, d_{\text{model}} \rceil$ MLP input channels by activation magnitude and zero the rest (weights remain unchanged). The quantization knob $q$ is implemented as **ZeroQuant-style** (Yao et al., 2022) activation quantization: we apply symmetric per-token fake-quantization at $q$ bits to the MLP output (dynamic range per token), with $q=16$ recovering the dense path. Full implementation details for all three axes are provided in Appendix E.

**GRPO with per-sequence budget sampling.** For each training input, we run a dense prefill on the context prefix to initialize the KV cache. We then create $K$ counterfactual compute schedules of length $T$ (the GRPO group size) that share the same teacher-forced token path; only the efficiency actions differ across schedules. This isolates the effect of compute allocation while keeping supervision fixed. To train a single policy that operates across a range of efficiency regimes, we *randomly sample* requested budget targets per sequence during training. Concretely, for each input we sample $(C_\kappa, C_\rho, C_\eta)$ from user-specified ranges (or discrete lists) and provide them to the policy as part of its observation. The same sampled targets are used for all $K$ schedules of that input and define the episode penalty $\mathcal{C}$. We use

process-level GRPO (Shao et al., 2024): at each time step $t$, we compute group-relative advantages by comparing the $K$ schedules for the same input (mean-centering across schedules at fixed $t$), and then apply a global whitening across all $(t, k)$ in the batch. Using these advantages, we update the policy with a clipped policy-gradient objective (PPO-style) and an entropy bonus for exploration.

## 5. Experiments

**Model families and naming.** We evaluate **Self-Optimizing Language Model (SOL)** controllers that allocate compute dynamically at each decode step, keeping the base LLM frozen. We first describe our naming convention, a model name encodes (i) *which efficiency axes are controlled*, (ii) the *granularity* of the action space and (iii) the control horizon (episode length):

$$\text{SOL-} \underbrace{\text{X}}_{\text{efficiency axes}} - \underbrace{\text{G}}_{\text{granularity}} - \underbrace{\text{T}k}_{\text{horizon}}.$$

We have three key parts here: **Axis tag X.** `Context (C)` controls token-level attention sparsity (keep-rate $\kappa$). `Quant (Q)` controls activation quantization (bit-width ratio $\eta$). `Prune (P)` controls structured MLP activation pruning (keep-rate $\rho$). `Joint (J)` controls all efficiency axes simultaneously. **Granularity tag G.** `2L` and `3L` denote *2* or *3* discrete levels per enabled axis (e.g., `SOL-J-2L-T16` has $2^3=8$ joint actions; `SOL-J-3L-T16` has $3^3=27$). `Fine (FL)` denotes a multi-level action space with many choices per axis (full definition in Appendix B). **Episode-length tag -Tk.** Models with a suffix `-Tk` vary the episode length (control horizon) $T=k$ while using a fixed mid-size joint action space (336 actions; Appendix B). For example, `SOL-J-FL-T4`, `SOL-J-FL-T16`, and `SOL-J-FL-T64` share the same action space but differ in how long the policy acts before a KV refresh; this horizon also determines the training episode length.

**Training procedure.** All controllers are trained with teacher-forced counterfactual compute schedules as described in Section 4. For each training input we run a dense prefill over a 1024-token prefix to fill the LLM KV cache, then decode an episode of $T$ decode steps (default $T=16$) in which the policy selects an efficiency action at each step. Within an episode, the token path is fixed (teacher forcing) and only the compute schedule varies across sampled schedules, allowing us to attribute differences in language-model loss to efficiency action allocation decisions instead of sampling noise of the LLM itself. We update the controller by sampling multiple schedules per input (GRPO group size $K$) and computing group-relative advantages. During training we sample requested budget targets per sequence and provide them to the policy; the reward combines token

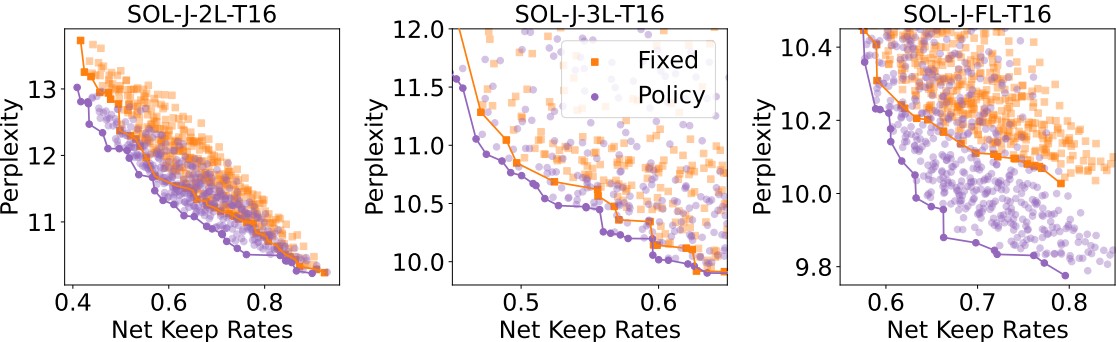

*Figure 2.* Quality–efficiency trade-offs for jointly-controlled SOL policies with increasingly fine-grained action sets: **(Left)** `SOL-J-2L-T16` (8 joint actions), **(Middle)** `SOL-J-3L-T16` (27 joint actions), and **(Right)** `SOL-J-FL-T16` (1560 joint actions).

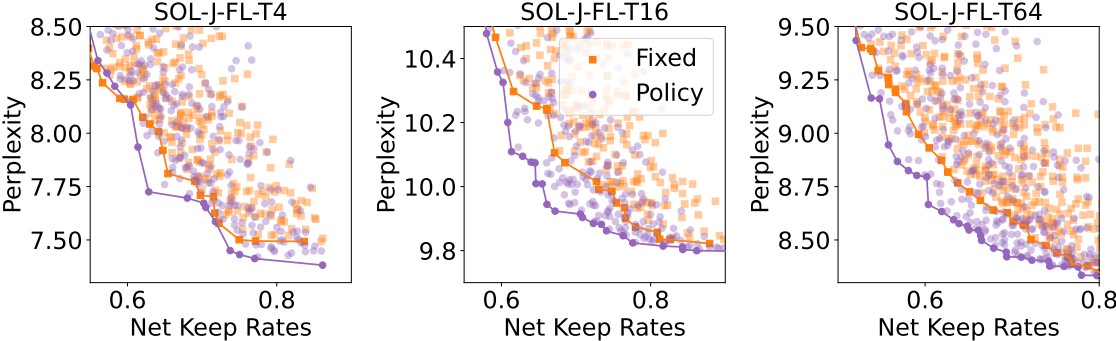

*Figure 3.* Effect of episode length $T$ (KV-refresh period) on the quality–efficiency frontier for a joint controller with a shared action space: **(Left)** `SOL-J-FL-T4`, **(Middle)** `SOL-J-FL-T16`, and **(Right)** `SOL-J-FL-T64`.

log-likelihood with appropriate penalties that encourage the *episode-average* budget usage to match the requested targets (Eqns. 2–4). The policy state is reset at episode boundaries, so inference-time overhead is constant per decoded token and negligible relative to the base LLM forward pass. Hyperparameters are reported in Appendix B.

We evaluate SOL by measuring language-model perplexity under the efficiency actions selected by the controller. For each evaluation input, we run a dense prefill on the first 1024 tokens and then teacher-force the next $T$ tokens while executing the controller's actions inside the frozen LLM. We report perplexity on this episode segment by averaging the negative log-likelihood over the $T$ evaluated positions (excluding the dense prefill), and exponentiating (giving us the perplexity). Unless stated otherwise, we select actions greedily (argmax) at evaluation time. Throughout, we treat keep-rates and normalized activation bit-width as monotonic proxies for compute and report quality versus the requested/realized budget targets rather than hardware-specific latency. Unless stated otherwise, evaluation uses $T{=}16$ and the same always-dense sink/window token conventions as in training; additional evaluation details and all action-space definitions appear in the appendix.

**Efficiency metric: net keep-rate.** Most experiments report quality against a scalar *net keep-rate*, which summarizes the realized episode-average resource usage across the enabled efficiency axes. For an episode, let $\bar{\kappa}$ be the realized token-attention keep-rate, $\bar{\rho}$ the realized MLP-channel keep-rate, and $\bar{\eta} = \bar{q}/16$ the realized normalized activation precision. When all three axes are enabled, we define

$$\text{NetKeep} = \frac{1}{3}\left(\bar{\kappa} + \bar{\rho} + \bar{\eta}\right).$$

If only a subset of axes is enabled, the average is taken over that subset. Lower net keep-rate means more aggressive compression. We use this metric as an architecture-agnostic proxy for retained compute capacity, not as a universal hardware metric: the exact latency, memory, and throughput gains depend on the kernels and hardware implementation.

**Action spaces.** Here, we ask whether a learned controller remains useful as the discrete search space grows. To test robustness to action-space complexity, we train three jointly-controlled policies with the same episode length ($T{=}16$) and the same three efficiency axes (token sparsity, MLP pruning, and activation quantization), but with increasingly fine-grained action sets: `SOL-J-2L-T16`,

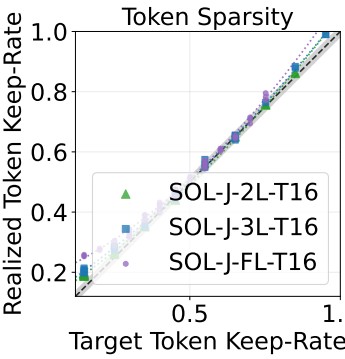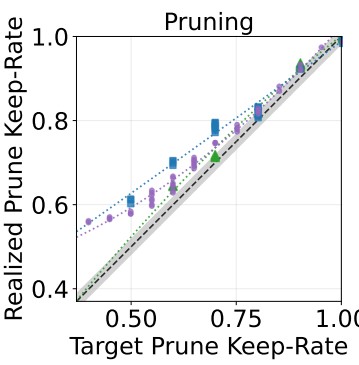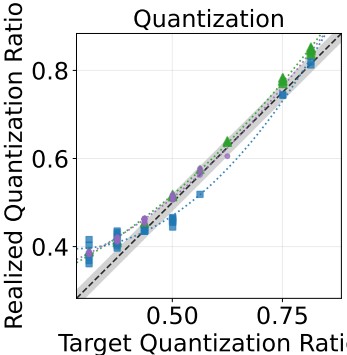

*Figure 4.* Budget adherence of SOL when provided a range of requested targets. The plots compare **(Left)** requested vs. realized token-attention keep-rate $\kappa$, **(Middle)** requested vs. realized structured MLP keep-rate $\rho$, and **(Right)** requested vs. realized normalized quantization ratio $\eta$.

`SOL-J-3L-T16`, and `SOL-J-FL-T16`. The 2L and 3L variants expose $2^3=8$ and $3^3=27$ joint actions respectively, while the fine-grained (FL) variant uses the Cartesian product of the per-axis choices (e.g., $|\mathcal{K}|\times|\mathcal{R}|\times|\mathcal{Q}| = 10\times13\times12 = 1560$ joint actions in our default setting; see Appendix B).

At evaluation time, we sweep a grid of requested budgets and measure the resulting quality–efficiency trade-off under teacher forcing. Specifically, we request token keep targets from 0.15–0.95, prune keep targets from 0.40–1.00, and quantization targets from 5–13 bits. To display these results, we collapse the three realized budgets into a single scalar *net keep-rate* proxy by averaging the realized token keep, prune keep, and normalized quantization ratio. We compare SOL against a *fixed* baseline that uses a static action schedule (constant over decode steps) and, when a requested budget falls between discrete action levels, mixes adjacent actions across the batch to match the requested average budget. Figure 2 shows that SOL consistently achieves lower perplexity than the fixed baseline at matched net keep-rate, and that these gains persist as the action space grows from 8 to 1560 options. Across operating points, SOL tracks the lower envelope of the random-schedule landscape and remains competitive with best-of-500 random search at matched net keep-rate (Appendix A, Figs. 8–9). Appendix C reports paired statistics over the full sweeps, including standard deviations, paired differences, significance tests, and policy win rates.

**Horizon** The length of the control horizon determines how long efficiency decisions can compound before the next KV refresh and policy-state reset. Longer horizons may be useful in deployment (e.g., to align with larger paging schemes or to reduce refresh overhead), but may also amplify effects such as KV-pollution and increase the difficulty of credit assignment. We therefore study whether

SOL remains effective as the episode length $T$ varies.

We train three joint policies that share the same joint action space (1560 actions; Appendix B) but differ in horizon: `SOL-J-FL-T4`, `SOL-J-FL-T16`, and `SOL-J-FL-T64`. In each case, the policy acts for $T$ decode steps and is then reset; the evaluation protocol matches training, using teacher-forced trajectories and the same always-dense sink/window convention. We evaluate all horizons on the same sweep of requested budgets as in the action-space study and report perplexity versus net keep-rate.

As shown in Figure 3, SOL improves the quality–efficiency trade-off over the fixed baseline across all horizons considered. While longer horizons have more risk of compounding error, the controller still learns policies that are on a better Pareto frontier, demonstrating that SOL can operate effectively under different KV-refresh schedules.

**Steering policy across efficiency targets** A key advantage of SOL is control over budgets at deployment: since the policy is conditioned on the requested budgets, we should be able to *specify* a target efficiency regime at inference time and have the policy hit the target without additional tuning. We evaluate this by measuring *budget adherence*: for a range of requested target triples $(C_\kappa, C_\rho, C_\eta)$, we run the policy under teacher forcing and compute the realized episode-average budgets $(\bar{\kappa}, \bar{\rho}, \bar{\eta})$ over effective steps.

Figure 4 plots requested versus realized budgets for operating points drawn from the policy's Pareto frontier (Figure 2). Overall, the controller tracks the requested targets closely: most points fall near the identity line, consistent with training under a budget-matching penalty with a $\tau=0.02$ tolerance band. Deviations arise because the budget penalty has finite weight, so under extremely aggressive settings the policy trades budget violations for large improvements in

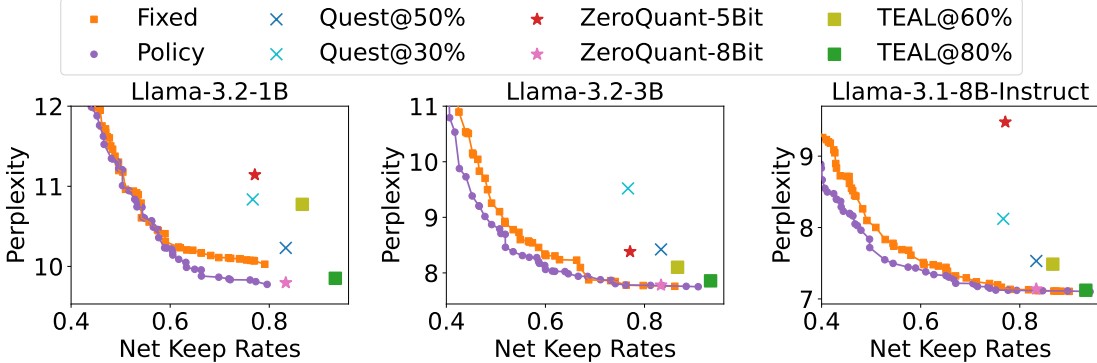

*Figure 5.* Training policies for a range of model sizes. Each subplot has policy configured to `SOL-J-FL-T16`, with varying LLM sizes.

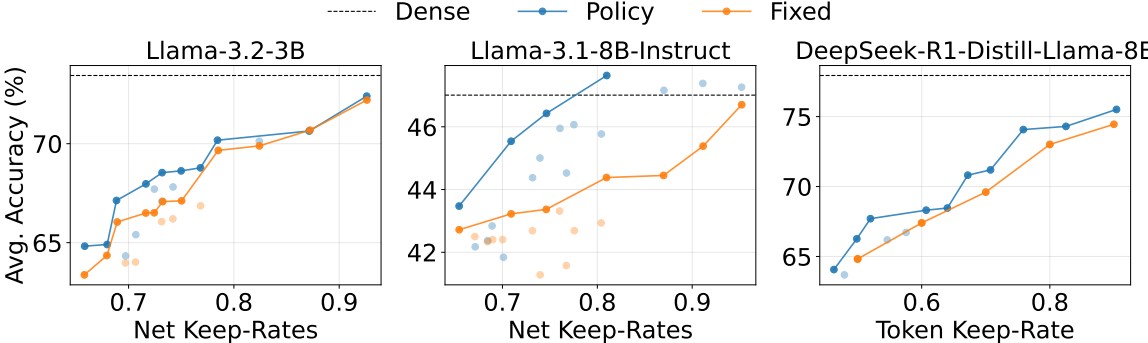

*Figure 6.* We evaluate policies against a fixed (static) compute schedule at several operating points (net keep-rate $\approx$ 0.5–0.9). **Left:** Average accuracy over `ARC-Easy`, `PIQA`, and `WinoGrande`. **Middle:** Average accuracy over three MMLU subjects in the continuation format. **Right:** Accuracy on `GSM8K-CoT-Llama` with 8-shot prompting with Token-Sparsity only (Quest page size 16).

language-model likelihood. We observe this most clearly for very low prune-keep requests (e.g., $\rho \lesssim 0.5$), where perplexity degrades sharply; in this regime the controller tends to choose less aggressive pruning than requested.

**Scaling to larger model sizes.** We keep the same policy architecture and train it on three base LLMs: `Llama-3.2-1B`, `Llama-3.2-3B`, and `Llama-3.1-8B-Instruct`, using an identical action space of 1560 discrete choices per decode step. For the 3B and 8B models we train with a prefill context length of 512 due to VRAM constraints. Figure 5 summarizes a large sweep over the efficiency search space; for readability we plot only the Pareto-optimal frontier. We show two frontiers: a *Fixed* baseline that uses a static, layer-agnostic efficiency setting throughout decoding, and our learned *Policy* that adaptively selects actions per step. To contextualize well-known static efficiency methods, we additionally annotate their corresponding operating points in the same plot: **Quest** token sparsity (Tang et al., 2024) at fixed keep rates (e.g., Quest@50% and Quest@30%), **ZeroQuant**-style activation quantization (Yao et al., 2022) at fixed bitwidths (ZeroQuant-8Bit and ZeroQuant-5Bit), and **TEAL**-style activation pruning (Liu et al., 2024a)

at fixed pruning levels (TEAL@80% and TEAL@60%). Concretely, each annotated point applies the named method at the indicated setting while holding the other efficiency knobs to their dense configuration, providing a direct reference for how each static method trades perplexity against net keep rate on each model. Across all three model sizes, the learned policy consistently discovers a better quality–efficiency Pareto frontier than fixed strategies, including those anchored at these standard static baselines.

**Downstream Evaluation.** We test whether SOL's per-token compute allocation improves downstream accuracy at matched budget. Concretely, we evaluate `SOL-J-FL-T16` controllers trained for `Llama-3.2-3B` and `Llama-3.1-8B-Instruct`: for `Llama-3.2-3B`, we report average accuracy over `ARC-Easy`, `PIQA`, and `WinoGrande` (Clark et al., 2018; Bisk et al., 2020; Sakaguchi et al., 2021), and for `Llama-3.1-8B-Instruct`, we report average accuracy over MMLU `conceptual_physics`, `high_school_chemistry`, and `international_law` (Hendrycks et al., 2021) in the continuation setting (Gao et al., 2024), where each candidate answer is scored as a textual continuation (via log-likelihood) and the prediction is the highest-likelihood

completion, reflecting a true completion-style evaluation rather than decoding letters for multi-choice questions. We sweep requested operating points with *net keep-rate* between 0.6 and 0.9 and compare against a *fixed* baseline that uses a static (per-step constant) efficiency action schedule matched to the same average budget. We also train a policy for the `DeepSeek-R1-Distill-Llama-8B` model for token sparsity using Quest with page size 16 and a horizon of 64 tokens before KV refresh. We evaluate the model on `GSM8K-CoT-Llama` with 8-shot prompting, where the model must decode both the reasoning trace and the final answer, stressing long-form text generation. Figure 6 shows that SOL consistently improves average accuracy over the fixed baseline at matched net keep-rate for all models, suggesting that the learned per-token allocation transfers beyond perplexity and improves end-task correctness under the same overall compute budget.

## 6. Discussion and Conclusion

Most inference-efficiency methods focus on how to compress an LLM's weights or activations to meet a target budget, choosing sparsity / quantization / pruning methods (block sparsity, channel pruning etc.) to deliver a single operating point. SOL instead learns *how much compute to spend per token*. A lightweight controller reads the model's activations and selects discrete efficiency actions, balancing model quality with our budget-matching objective to meet efficiency requirements. While we use monotonic proxies for compute (keep-rate for tokens/channels/bits), our approach could leverage the surplus of efficiency data (real latency, power) that such policies can optimize for. SOL opens an orthogonal axis of efficiency optimization: learning a policy that adapts compute allocation to the difficulty of the generation process and constraints of the serving environment in which the LLM is deployed.

## Acknowledgments

This work was supported in part by the NSF CAREER Grant No. 2339084 and by an Intel Labs Research Gift. We would like to thank Xingyou Song for providing feedback on the initial drafts of this paper.

## Impact Statement

This paper presents work whose goal is to advance the field of Machine Learning. There are many potential societal consequences of our work, none that we feel must be specifically highlighted here.

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

# A. Appendix

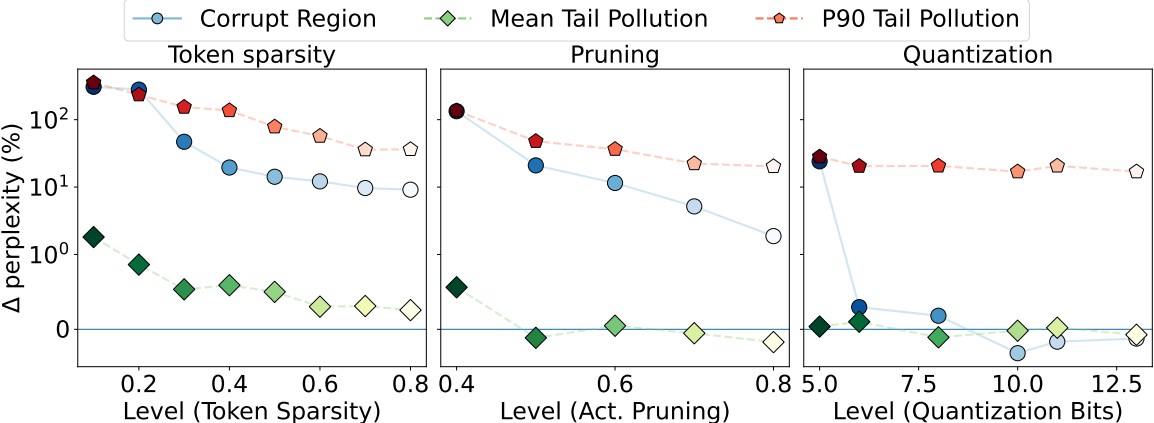

*Figure 7.* Axis-dependent KV-pollution: mean direct $\Delta$ perplexity on the corrupted steps vs. P90 peak $\Delta$ perplexity in the subsequent dense tail.

**KV Pollution.** We quantify KV-pollution using teacher-forced `WikiText` windows (Figure 7). For each window, we run a dense prefill on a 512-token prefix, and decode the next $t_{\text{corrupt}} = 4$ tokens using *one* efficiency mechanism: token sparsity (keep fraction $\kappa$), structured MLP activation pruning (keep fraction $\rho$) or activation quantization (bit-width $q$), and then switch back to fully dense decoding ($\kappa=1, \rho=1, q=16$) for the remainder of the trajectory. We compare this run to an all-dense baseline under the same token path and report $\Delta$ perplexity as percent change, $\Delta\text{ppl}(\%) = (\exp(\Delta\text{NLL})-1)\cdot 100$. Figure 7 shows three plots, where we measure the $\Delta$ perplexity as a percentage change in the corrupt region itself (where the optimization was active), and then the impact of the corrupt region on the subsequent tokens after returning to dense compute. This *tail pollution* isolates delayed errors caused by the polluted KV states. While the mean tail pollution is low, the P90 tail pollution remains higher even than the perplexity change in the corrupt region. Here, we can also see that token sparsity exhibits the highest KV-pollution, which is expected, as missing tokens directly impact the information that is written to the KV-cache.

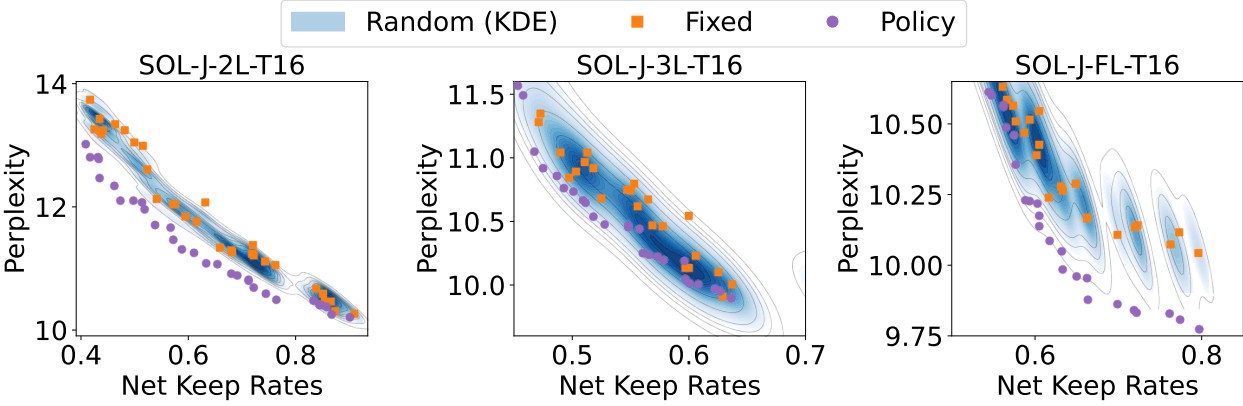

*Figure 8.* For each model variant (`SOL-J-2L-T16`, `SOL-J-3L-T16`, `SOL-J-FL-T16`), we sample random counterfactual compute schedules at Pareto operating points and visualize the resulting distribution as a 2D KDE over (net keep-rate, perplexity). We overlay the fixed (static) allocation baseline and the learned policy.

**Random-search landscape at matched budgets.** We contextualize the learned controller against the distribution demonstrated by uninformed compute schedules. For each model in Figure 2, we take the Pareto-frontier configurations produced by the policy (each corresponding to a particular budget regime) and, for each configuration, sample 30 random counterfactual compute schedules by uniformly drawing actions from the same discrete action set (8, 27, and 1560 actions for `2L`, `3L`, and

`FL`). We evaluate each schedule under teacher forcing and record its realized net keep-rate (average of token, channel, and quantization keep-rates) and resulting perplexity. Across model variants, the policy consistently lies on the lower envelope of the random density cloud and improves over the fixed baseline, indicating that SOL is not simply matching a typical random schedule, but selecting compute allocations that achieve lower perplexity at comparable net keep-rates. Pooling these random schedules across all Pareto points yields a dense set of samples, which we visualize as a 2D KDE over (net keep-rate, perplexity) in Figure 8. We overlay the fixed baseline (constant action across decode steps) and the policy.

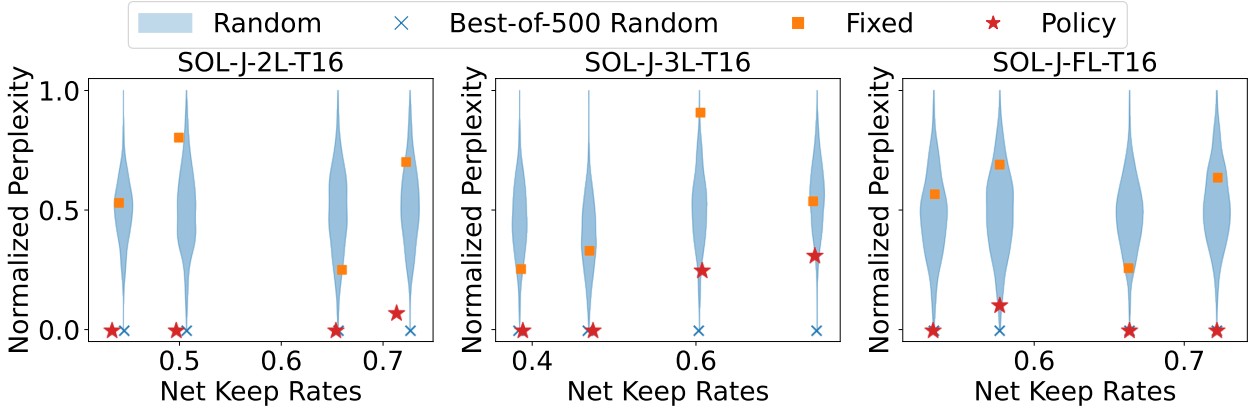

*Figure 9.* Comparison against 500-sample random schedule search. For selected Pareto operating points in `SOL-J-2L-T16`, `SOL-J-3L-T16`, and `SOL-J-FL-T16`, violins show the distribution of normalized perplexity across 500 randomly sampled compute schedules at similar net keep-rates; markers denote the fixed baseline, the best-of-500 random schedule, and the learned policy. Lower is better.

**Policy vs. random search.** While the KDE view summarizes the global landscape, it does not directly answer how the policy compares to *strong* random search at a specific operating point. We therefore run a targeted stress test: for each of `SOL-J-2L-T16`, `SOL-J-3L-T16`, and `SOL-J-FL-T16`, we sample four Pareto-frontier configurations from Figure 2 and, for each, evaluate 500 random counterfactual schedules (again sampling actions uniformly from the same action set), alongside the fixed baseline and the policy. Figure 9 reports the resulting per-point random distributions (violins) together with the fixed and policy perplexities (with row-wise normalization for comparability across operating points). Across the 12 test points, the policy matches or exceeds the best-of-500 random schedule in 8 cases. In the remaining 4 cases, the policy remains in the extreme tail of the random distribution (top $5.6\%$ at worst; i.e., at most 28 out of 500 random schedules outperform it), and its regret relative to the best random schedule is small (maximum gap $0.085$ perplexity points). Together with Figure 8, these results show that SOL learns a non-trivial per-token allocation strategy: it tracks the lower envelope of the random-search landscape globally, and remains competitive even against substantial pointwise random search over the same efficiency mechanisms.

**Designing static (non-learned) strategies for budget allocation.** Beyond fixed and random schedules, we evaluate two simple *hand-crafted* criteria that allocate per-token compute using a scalar signal from the *previous* decoding step, while still steering the *episode-average* budgets to match the requested targets. **Entropy-Matched Criterion (EMC).** We compute an uncertainty score $u_{t-1} \in [0, 1]$ from the previous step's logits using normalized entropy $\hat{H}_{t-1}$ (Xin et al., 2020); higher uncertainty biases the next action toward *less aggressive* compression (larger attention keep-rate $\kappa$, higher MLP keep-rate $\rho$, and/or higher activation precision $\eta$). **Drift-Aware Criterion (DAC).** We compute a representation-change score $d_{t-1} \in [0, 1]$ as the cosine drift between consecutive last-layer hidden states, $d_{t-1} = \frac{1}{2}(1 - \cos(h_{t-1}, h_{t-2}))$ (using embedding drift for the first decoded token) (Schuster et al., 2022), and allocate more compute on high-drift steps. For both baselines, actions are chosen from the same discrete action set as SOL using a greedy *budget-steering* rule: at each step and for each enabled axis, we select between the two nearest discrete levels around the remaining required average, then a feasible action so the target budget remain attainable over the remaining steps. From Figure 10, we find that these hand-crafted static strategies do not outperform the policy. Jointly trading off different optimization methods allow us to hit significantly more aggressive efficiency targets than hand-designed strategies for individual optimization methods.

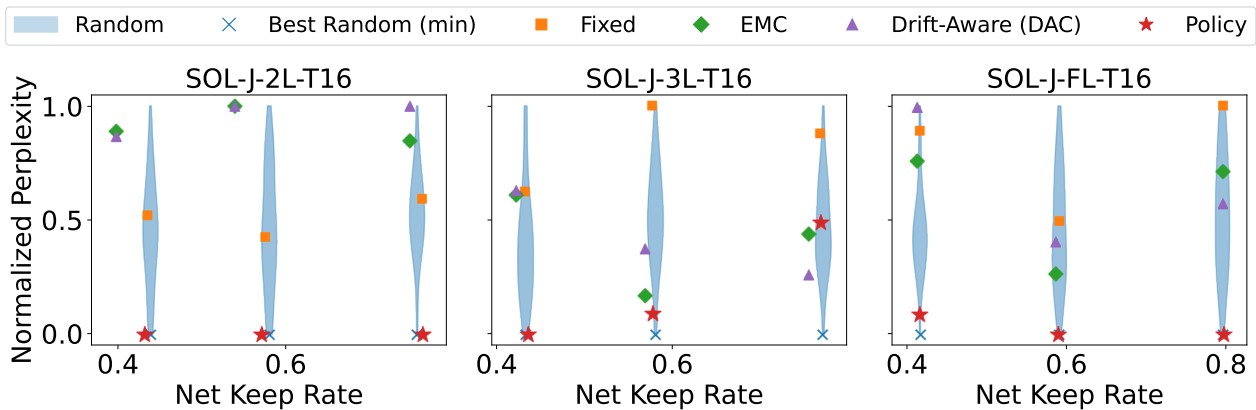

Figure 10. Hand-crafted budget-allocation heuristics (entropy- and drift-based) do not consistently improve over the fixed baseline, while the learned SOL policy achieves the best quality–efficiency trade-off.

**Optimizing individual efficiency methods.** To isolate whether SOL's gains come from *joint* multi-axis control or from learning a per-token controller in the simplest setting, we train three *single-axis* controllers for each efficiency method: `SOL-Q-2L-T16` (quantization only), `SOL-P-2L-T16` (MLP pruning only), and `SOL-C-2L-T16` (token sparsity only). In all cases, the episode length is $T{=}16$ and the action space contains only two levels for the active axis; the other axes are held fixed at their dense settings. We evaluate three binary action sets per axis (x-axis labels in Figure 11), corresponding to different choices of the two available levels (e.g., two candidate bit-widths for quantization).

For each binary setting, we compare the learned controller against (i) a *fixed* baseline that uses a static schedule over the same two levels and (ii) *random* compute schedules that sample one of the two actions at each decode step. We report *normalized* perplexity for each setting (normalized within each setting for comparability across axes). As shown in Figure 11, SOL consistently selects schedules that lie on the lower envelope of the random-search landscape, and it matches the *best-of-30* random schedule in 7 out of the 9 binary settings tested. This indicates that even with a minimal two-level action space, SOL learns a non-trivial per-token allocation strategy beyond what is obtained by static or uninformed schedules.

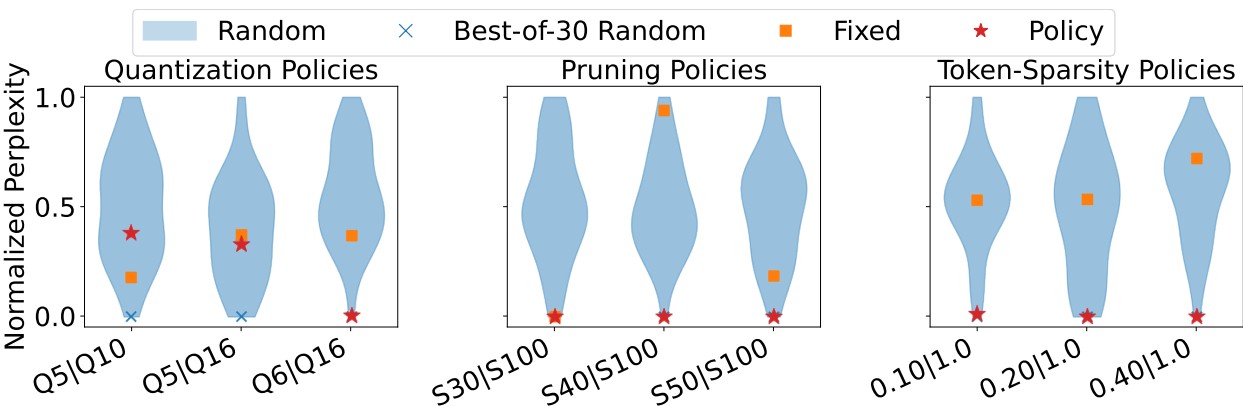

Figure 11. Per-axis optimization with two-level action sets. Each subplot trains three single-axis controllers (`SOL-Q/P/C-2L-T16`). For each two-level action set (x-axis), the violin shows the distribution of *normalized* perplexity over 30 randomly sampled schedules. Lower is better.

# B. Model Configurations

We primarily train and evaluate six main models, whose configurations are described below.

## B.1. Search Space Size Ablation Models
### Shared configuration (used unless overridden)

**Base LLM:** `meta-llama/Llama-3.2-1B`

**Token Sparsity Implementation:** Quest (page size $= 4$)

**Budget Ranges:** Token budget $[0.1, 1.0]$; pruning budget $[0.4, 1.0]$; quantization ratio budget $[0.3125, 1.0]$

**Budget Penalty Weights:** $\alpha_\kappa = 100$; $\alpha_\rho = 100$; $\alpha_\eta = 200$

**GRPO group size / horizon:** $T{=}16$ decode steps/episode; $K{=}16$ schedules/input; entropy coef $= 0.05$

**Optimization:** Batch size $= 8$; grad accumulation $= 8$; lr $= 10^{-4}$; max grad norm $= 2.0$; epochs $= 1$

**Context Length:** 1024

**Policy Network (Controller):** Transformer: $d_{\text{model}} = 512$; $n_{\text{heads}} = 4$; $n_{\text{layers}} = 1$; MLP ratio $= 4.0$; action dim $= 32$; dropout $= 0.0$

**PPO Settings:** PPO clip $= 0.2$; PPO epochs $= 1$; minibatch $= 512$; target batch $= 2048$

**Sink and Window Tokens:** Sink tokens $T_s = 4$; window tokens $T_w = 2$

**Dataset:** `allenai/dolma` (`v1_6-sample`); text field: `text`; dataset pct: 100%; seed: 1234

### Binary: SOL-J-2L-T16 (8 choices per decode step)

**Shared settings:** See shared config in B.1.

**Action Space:** **Keep-Rate** $\{0.1, 1.0\}$; **Prune** $\{\text{s60, s100}\}$; **Quant** $\{\text{q5, q16}\}$

**Override: Budget Ranges:** Pruning budget $[0.6, 1.0]$ (token/quant as shared)

### Ternary: SOL-J-3L-T16 (27 choices per decode step)

**Shared settings:** See shared config in B.1.

**Action Space:** **Keep-Rate** $\{0.1, 0.6, 1.0\}$; **Prune** $\{\text{s40, s80, s100}\}$; **Quant** $\{\text{q5, q7, q16}\}$

### Multi: SOL-J-FL-T16 (1560 choices per decode step)

**Shared settings:** See shared config in B.1.

**Override: Token Sparsity Implementation:** Quest (page size $= 8$)

**Override: Group size:** $K{=}32$ schedules/input ($T$/entropy as shared)

**Action Space:** **Keep-Rate** $\{0.1, 0.2, ..., 1.0\}$; **Prune** $\{\text{s40, s45, ..., s100}\}$; **Quant** $\{\text{q5, q6, ..., q16}\}$

## B.2. Horizon (Episode length) ablation
### Shared configuration (used unless overridden)

**Base LLM:** `meta-llama/Llama-3.2-1B`

**Token Sparsity Implementation:** Quest (page size $= 8$)

**Action Space:** **Keep-Rate** $\{0.1, 0.4, 0.7, 1.0\}$; **Prune** $\{\text{s45, s50, s60, s70, s80, s90, s100}\}$; **Quant** $\{\text{q5, q6, ..., q16}\}$

**Budget Ranges:** Token budget $[0.1, 1.0]$; pruning budget $[0.45, 1.0]$; quantization ratio budget $[0.3125, 1.0]$

**Budget Penalty Weights:** $\alpha_\kappa = 100$; $\alpha_\rho = 100$; $\alpha_\eta = 200$

**GRPO group size / horizon (default):** $T$=16 decode steps/episode; $K$=32 schedules/input; entropy coef = 0.05

**Optimization:** Batch size = 8; grad accumulation = 8; lr = $10^{-4}$; max grad norm = 2.0; epochs = 1

**Context Length:** 1024

**Policy Network (Controller):** Transformer: $d_{\text{model}} = 512$; $n_{\text{heads}} = 4$; $n_{\text{layers}} = 1$; MLP ratio = 4.0; action dim = 32; dropout = 0.0

**PPO Settings:** PPO clip = 0.2; PPO epochs = 1; minibatch = 512; target batch = 2048

**Sink and Window Tokens:** Sink tokens $T_s = 4$; window tokens $T_w = 2$

**Dataset:** `allenai/dolma` (`v1_6-sample`); text field: `text`; dataset pct: 100%; seed: 1234

**Horizon 4: SOL-J-FL-T4**

**Shared settings:** See shared config in B.2.

**Override: Horizon:** $T$=4 decode steps/episode ($K$/entropy as shared)

**Horizon 16: SOL-J-FL-T16**

**Shared settings:** See shared config in B.2.

**Horizon 64: SOL-J-FL-T64**

**Shared settings:** See shared config in B.2.

**Override: Horizon:** $T$=64 decode steps/episode ($K$/entropy as shared)

**B.3. Model Scaling Experiments**
**meta-llama/Llama-3.2-3B**

**Shared Settings:** See shared config in B.1.

**Base LLM:** `meta-llama/Llama-3.2-3B`

**Override: Context Length:** 512

**Override: Optimization:** Batch size = 2; grad accumulation = 32 (lr/max grad norm/epochs as shared)

**Override: GRPO group size / horizon:** $T$=16 decode steps/episode; $K$=16 schedules/input; entropy coef = 0.05

**Override: Policy Network (Controller):** Same as shared, except max length = 512

**meta-llama/Llama-3.1-8B-Instruct**

**Shared Settings:** See shared config in B.3.

**Base LLM:** `meta-llama/Llama-3.1-8B-Instruct`

**Override: Context Length:** 512

**Override: Optimization:** Batch size = 2; grad accumulation = 32 (lr/max grad norm/epochs as shared)

**Override: GRPO group size / horizon:** $T$=16 decode steps/episode; $K$=16 schedules/input; entropy coef = 0.05

**Override: Policy Network (Controller):** Same as shared, except max length = 512

**deepseek-ai/DeepSeek-R1-Distill-Llama-8B**

**Shared Settings:** See shared config in B.1.

**Base LLM:** `deepseek-ai/DeepSeek-R1-Distill-Llama-8B`

**Override: Token Sparsity Implementation:** Quest (page size = 16)

**Action Space: Keep-Rate** `{0.05, 0.1, 0.2, 0.3, 0.4, 0.5, 0.6, 0.7, 0.8, 0.9, 1.0}`; **Prune** `{s100}`; **Quant** `{q16}`

**Override: Budget Ranges:** Token budget $[0.1, 1.0]$; pruning budget fixed at 1.0; quantization ratio budget fixed at 1.0

**Override: Budget Penalty Weights:** $\alpha_\kappa = 100$; $\alpha_\rho = 0$; $\alpha_\eta = 0$

**Override: GRPO group size / horizon:** $T=64$ decode steps/episode; $K=16$ schedules/input; entropy coef $= 0.05$

**Override: Optimization:** Batch size $= 2$; grad accumulation $= 32$; lr $= 10^{-4}$; max grad norm $= 2.0$; epochs $= 1$

**Override: Sink and Window Tokens:** Sink tokens $T_s = 16$; window tokens $T_w = 16$

**Override: Policy Network (Controller):** Same as shared, except max length $= 512$

## C. Statistical significance of policy gains

For each evaluated configuration, we compare SOL and the fixed baseline at matched requested budget targets. Each row in Table 1 aggregates over the corresponding budget sweep. We report mean perplexity, the paired difference $\Delta = \text{PPL}_{\text{policy}} - \text{PPL}_{\text{fixed}}$, a one-sided paired $t$-test, and the fraction of target configurations for which the policy achieves lower perplexity. Negative $\Delta$ indicates that SOL is better.

*Table 1.* Paired comparison between SOL and fixed allocation over matched budget sweeps. SOL achieves significantly lower perplexity across all configurations.

| Config | Policy PPL | Fixed PPL | $\Delta$ | $p$-value | Win rate |
|---|---|---|---|---|---|
| `SOL-J-FL-T4` | $9.53 \pm 1.87$ | $9.66 \pm 2.20$ | $-0.127 \pm 0.748$ | $3.2 \times 10^{-5}$ | 58.7% |
| `SOL-J-FL-T16` | $11.26 \pm 1.29$ | $11.42 \pm 1.60$ | $-0.160 \pm 0.696$ | $3.0 \times 10^{-8}$ | 70.4% |
| `SOL-J-FL-T64` | $9.12 \pm 0.64$ | $9.34 \pm 0.79$ | $-0.212 \pm 0.259$ | $< 10^{-10}$ | 84.7% |
| `SOL-J-2L-T16` | $11.51 \pm 0.60$ | $11.89 \pm 0.72$ | $-0.383 \pm 0.241$ | $< 10^{-10}$ | 95.3% |
| `SOL-J-3L-T16` | $10.63 \pm 0.77$ | $10.94 \pm 1.08$ | $-0.309 \pm 0.346$ | $< 10^{-10}$ | 84.4% |
| `Llama-3.2-3B` | $9.14 \pm 1.19$ | $9.46 \pm 1.28$ | $-0.327 \pm 0.243$ | $< 10^{-10}$ | 95.5% |
| `Llama-3.1-8B` | $7.93 \pm 0.62$ | $8.20 \pm 0.77$ | $-0.270 \pm 0.240$ | $< 10^{-10}$ | 88.9% |

## D. Controller training cost

Table 2 reports approximate training cost for the SOL controller. The base LLM is frozen in all cases; only the lightweight policy network is trained.

*Table 2.* Approximate controller training cost on H100 GPUs.

| Base model | Training cost |
|---|---|
| `Llama-3.2-1B` | 4 GPU-hours |
| `Llama-3.2-3B` | 7 GPU-hours |
| `Llama-3.1-8B-Instruct` | 20 GPU-hours |

## E. Implementation details of efficiency actions

SOL actions select a discrete tuple $(\kappa, \rho, q)$ at each decode step, where $\kappa$ controls token sparsity in attention (Quest), $\rho$ controls structured activation pruning in the MLP, and $q$ controls activation quantization bit-width. We implement all three

knobs as *inference-time* controls in a frozen HuggingFace LLaMA model by monkey-patching the attention and MLP forward paths; the base model weights are unchanged.

### E.1. Token sparsity via Quest (context keep-rate $\kappa$)

**Budgeted token selection.**  At decode step $t$, let $K_t$ denote the number of keys available in the KV cache for the current query (i.e., the current KV length). Given a requested keep-rate $\kappa_t \in [0, 1]$, we convert it to a per-sequence *token budget*

$$b_t = \lceil \kappa_t \cdot K_t \rceil,$$

clamped to $[0, K_t]$. This budget is set per sequence in the batch and broadcast across heads.

**Quest masking in LLaMA attention.**  We implement token sparsity using Quest (Tang et al., 2024) by modifying the LLaMA `eager_attention_forward` path to add an *additive* mask (bias) before softmax. Concretely, we group the $K_t$ keys into contiguous pages of size $S$ (our `quest_page_size`), pad to a multiple of $S$, and compute a page-level upper bound score for each head and query. Let $q \in \mathbb{R}^d$ denote the query vector and let a page contain keys $\{k^{(j)}\}_{j=1}^{S}$. Define per-dimension page extrema:

$$m^+ = \max_j k^{(j)}, \qquad m^- = -\min_j k^{(j)}.$$

We decompose the query into positive and negative magnitudes, $q_{\text{pos}} = |q| \odot \mathbf{1}[q \geq 0]$ and $q_{\text{neg}} = |q| - q_{\text{pos}}$, and score pages by the Quest bound:

$$\text{score}(\text{page}) = q_{\text{pos}}^\top m^+ + q_{\text{neg}}^\top m^-.$$

We then keep the top $\lceil b_t/S \rceil$ pages according to this score, restricted to keys that are allowed by the model's attention mask (causal/padding). Expanding the selected pages yields a boolean token keep-mask of shape $[B, H, Q, K_t]$, which we convert to an additive bias:

$$\Delta M_t(i, h, q, k) = \begin{cases} 0 & \text{if key } k \text{ is kept} \\ -\infty & \text{otherwise.} \end{cases}$$

This bias is added to the existing attention mask and the original attention computation is left unchanged. If the requested budget keeps all allowed tokens, we bypass masking and execute dense attention.

**Granularity.**  Although $\kappa_t$ is specified per sequence, selection is performed per head (and per query position if $Q > 1$). In our autoregressive decoding setting, $Q = 1$ and the policy changes $\kappa_t$ once per token.

### E.2. Structured MLP activation pruning (keep-rate $\rho$)

**What is pruned.**  We implement structured pruning as *input-channel gating* on the residual stream entering each MLP block (HF `LlamaMLP`). This is a structured (channel-wise) activation mask applied at inference time; model weights remain unchanged.

**Per-token channel selection.**  Let $x_t \in \mathbb{R}^{d_{\text{model}}}$ be the MLP input activation for the token at step $t$ (teacher-forced decoding uses a single-token forward pass, so sequence length within the call is 1). Given a keep-rate $\rho_t \in [0, 1]$, we keep

$$m_t = \lceil \rho_t \cdot d_{\text{model}} \rceil$$

channels per example. We score channels by magnitude (max absolute activation within the current call):

$$s(j) = \max_{\text{token in call}} |x(j)|,$$

and keep the top-$m_t$ channels. The resulting binary mask is shared across tokens inside the call (and is therefore per-token in decode), and we apply it multiplicatively:

$$\tilde{x}_t = x_t \odot \mathbf{m}_t.$$

We apply pruning only inside the MLP block (i.e., we do *not* prune the attention input for the same layer), so attention projections remain dense for that layer while the MLP update is compressed.

**E.3. Activation quantization (bit-width $q$)**

**What is quantized.** We implement activation quantization as a *fake-quantization* operator applied to the *output* of each MLP block (the residual-stream update). The MLP matmuls are executed in full precision and the output is quantized/dequantized before being added back through the residual path.

**Quantizer.** Given MLP output activation $z \in \mathbb{R}^{d_{\text{model}}}$ and a selected bit-width $q \in \{5, \ldots, 16\}$, we use symmetric uniform quantization with per-token dynamic range. Let $q_{\text{max}} = 2^{q-1} - 1$ and let $a = \max_j |z(j)|$ (computed per token). The scale is $s = a/q_{\text{max}}$ and the quantized output is

$$\hat{z}(j) \ = \ \text{clip}\left(\left\lfloor \frac{z(j)}{s} \right\rceil, -q_{\text{max}}, q_{\text{max}}\right) \cdot s.$$

For $q \geq 16$ we return $z$ unchanged. Mixed-bit batches are supported by applying this operator per example conditioned on the selected $q$.

**Normalized precision ratio.** In the main text we use the normalized ratio $\eta = q/16 \in [0, 1]$ as the policy-controlled precision knob and as the target-matching quantity in the budget penalty.

