# OpenReview forum: "Compute Where it Counts: Self Optimizing Language Models"
_ICML.cc/2026/Conference — ICML 2026 regular_

### Official Review · Reviewer_XsE9 · 2026-02-21

**Soundness:** 3
**Presentation:** 2
**Significance:** 2
**Originality:** 3
**Overall Recommendation:** 4
**Confidence:** 3

**Summary:**

This works proposes an approach for dynamic budget allocation (token pruning, structured sparsity, activation quantization) via learning a small policy network. The policy network operates on episodes of fixed length = T and is trained to allocate an optimal configuration for a desired compression ratio. After the end of the episode, the cache is refreshed via dense forward pass. The introduced approach is validated on a compression of several language models from Llama family.

**Compliance With Llm Reviewing Policy:**

Affirmed.

**Final Justification:**

After reading the initial version of the paper, I had concerns about its practical applicability and questions related to the analysis. The authors have addressed these points, so I decided to increase the score to Weak Accept.

**Key Questions For Authors:**

- I would be illustrative to show some representative configurations found by the policy - i.e. the values for Quest/TEAL sparsity / precision.
Can one achieve practical speed-ups using the mixed-precision policy. What is the overhead of invoking policy network?
- It would be insightful to illustrate the chosen actions on a short sequence to see which tokens are treated as important/unimportant.

**Limitations:**

-

**Strengths And Weaknesses:**

Strengths

- The idea to learn a policy network to produce an optimal compressed configuration is sound and appears to be novel in the literature in the provided formulation.
- Policy-produced configuration seem to consistently outperform fixed precision / dropping rate configurations.

Weaknesses
- The search space seems to be limited in some directions. For instance, typically one is interested in quantization into 4 bits and below.
- The name of the paper is confusing. One would think that the language model is optimizing itself, whereas a separate network is trained.
- The submission code is not provided.

---

> ### Author Rebuttal · Authors · 2026-03-30
>
> Thank you for your review and feedback. Please find our responses below:
>
> > practical speed-ups … quantization into 4 bits and below.
>
> Please note that the quantization method itself is not the core focus of this work, our focus is on building a framework to enable LLMs to optimize their own decode process. We thus use relatively simple magnitude-based pruning, page-based token sparsity and quantization and demonstrate a clear trade-off in static methods when compared to our dynamic policy. As further detailed in our response to Reviewer 4B27:
>
> >> _More broadly, our goal is to establish the algorithmic value of dynamic per-token budget allocation across a unified action space. Some operating points we study, such as per-step mixed low-bit activation quantization combined with pruning and token sparsity do not have mature deployable kernels. Thus, hardware-specific latency numbers will conflate our proposed control framework. We will revise the paper to make this more explicit, and highlight end-to-end kernel-aware evaluation and reward formulations as important future work._
>
> > What is the overhead of invoking policy network?
>
> Minimal, Llama8B benchmark: prefill 1024 tokens decoding 512 tokens: 16.83 seconds vs 17.34 seconds. (Note that we do not use accelerated kernels since we study very fine grained pruning, quantization and token-sparsity)
>
> > name of the paper is confusing.
>
> We apologize for this confusion, technically for an LLM to optimize itself, we will need to map hidden states to an action set via a neural network. The alternative is to have reserved tokens for these actions, but it is prohibitively expensive to fine-tune an entire LLM to learn to map hidden states to efficiency actions through RL, and can cause the model weights to significantly change, forgetting its own instruction tuning (and introducing several confounding factors). We thus introduce a small predictor to leverage the LLM states without risking regression in the base model quality. Since the models' own hidden states effectively map to efficiency actions, we call it self optimizing. We can further discuss this in the introduction and abstract for clarity.
>
> > The submission code is not provided
>
> Please find our anonymized code here: https://anonymous.4open.science/r/SOL-73E4/README.md
> We will also release all models that we trained upon acceptance.
>
> >  illustrate the chosen actions on a short sequence
>
> We will definitely include more analysis on actions taken by the model and as well as representative configurations in the main paper.
>
> To study this within the scope of the rebuttal character limits, we analyze the per-step actions chosen by the SOL policy on Llama-3.1-8B-Instruct across three budget targets on a GSM8K math reasoning prompt. All three configurations produce the correct answer (72). TEAL and ZeroQuant are incorrect in the aggressive setting, Quest is wrong in both aggressive and moderate.
>
> The policy precisely tracks multi-axis targets:
>
> | Setting    | Target (κ, ρ, q) | Achieved (κ, ρ, q) | Correct |
> |-|-|-|-|
> | Aggressive | 0.30, 0.60, 6-bit | 0.29, 0.61, 6.0-bit | ✓ |
> | Moderate   | 0.50, 0.70, 8-bit | 0.49, 0.73, 7.1-bit | ✓ |
> | Mild       | 0.80, 0.90, 14-bit | 0.83, 0.93, 14.9-bit | ✓ |
>
> Dynamic action mixing: The policy does not adopt a fixed operating point. In moderate mode it uses 31 distinct (κ,ρ,q) combinations and changes κ at 78.5% of steps. A fixed Quest@50% uses exactly 1 combination for every step and fails in two out of the three settings.
>
> Example: periodic full-attention spikes. Under the aggressive budget (κ_avg=0.29), the policy uses heavy sparsity (κ=0.1–0.3) for most tokens but spikes to κ=1.0 approximately every 8 steps (median gap = 8, half the episode length T=16). These spikes consistently occur at lightweight tokens like "of" and "number" that are followed by denser reasoning sequences. For example, in the generated sequence "...the number of clips is the number of clips she sold in..." the bold tokens "number" receives κ=1.0 (full attention over the entire KV cache) while all surrounding tokens like "clips", "the", "in", "sold" receive κ=0.1–0.3 (only 10–30% of KV tokens attended). This is a budget-aware refresh strategy that no static method can replicate: Quest@29% would apply 29% uniformly to every step.
>
> Cross-axis trade-offs. In the aggressive setting the policy locks quantization at 6-bit for virtually all steps (the easiest axis to compress at this budget) but dynamically varies κ∈{0.1,...,1.0} and ρ∈{0.50,...,0.85}. In the mild setting, the opposite: the policy stays near-dense (κ=1.0 for 72% of steps, ρ≥0.95, q≥15-bit) but drops to κ=0.2–0.3 on predictable tokens like "clips" and sentence-ending punctuation, getting free compression where full context is unnecessary.
>
> These results show that SOL's policy learns qualitatively different behavior from any static baseline, dynamically concentrating compute on the tokens and axes where it matters most.

---

> > ### Author Rebuttal · Reviewer_XsE9 · 2026-04-01
> >
> > The concerns were partially resolved.
> >
> > However, the practicality of the proposed method is still questionable as the determined configurations may not yield speed-ups.
> >
> > **UPD** The results provided below provide the evidence of the practical applicability of the method. Therefore, I decide to raise the score.

---

> > > ### Author Response · Authors · 2026-04-01
> > >
> > > We believe we can resolve your remaining concern in a short rebuttal below. We believe these do not require a significant update to the paper, as token-sparsity acceleration methods are already well known, and our paper focuses on dynamic budget allocation strategies, not SoTA downstream latency. We show clear model-quality wins in dynamic settings, when compared to static budget allocation. We would really appreciate if you could take our response below into consideration.
> > >
> > > We have without any doubt shown that dynamic configurations are significantly stronger than static configurations in perplexity and downstream measurements. (Table in response https://openreview.net/forum?id=1OPGcMM2Xt&noteId=OzHFIovwn3)
> > >
> > > **To further provide evidence, we explicitly profile with Quest (page based sparsity), where downstream performance measurements can be made. We present below clear practical evidence of dynamic methods working while yielding speed-ups**
> > >
> > > ### With Quest sparse attention, lower keep-rates give clear speed-ups
> > >
> > > We obtain the following end-to-end speedups when using the Quest sparsity framework with our predictor (We profile **Llama-3.1-8B-Instruct** on **H200 140GB, FP16**):
> > >
> > > | Keep Fraction ($\kappa$) | Speedup @1K | Speedup @8K | Speedup @32K |
> > > |:-:|:-:|:-:|:-:|
> > > | 0.2 | 1.10x | 1.39x | 1.85x |
> > > | 0.3 | 1.07x | 1.32x | 1.66x |
> > > | 0.4 | 1.05x | 1.25x | 1.50x |
> > > | 0.5 | 1.04x | 1.19x | 1.37x |
> > >
> > > So for the exact sparse attention mechanism used by Quest, **the operating points selected by SOL are absolutely in the range where practical speed-ups are realized**.
> > >
> > > ### The policy network overhead is very small
> > >
> > > We directly measure the controller overhead, this time **with the Quest framework**, which is a reliable deployment friendly measurement.
> > >
> > > | Measurement | Value |
> > > |:--|:-:|
> > > | Policy.step() latency | 0.38 ms |
> > > | Policy parameters | 8.56M (0.11% of LM) |
> > > | Measured end-to-end overhead (episode_len = 16) | +3.9% |
> > > | Measured end-to-end overhead (episode_len = 64) | +3.5% |
> > > | Measured end-to-end overhead (episode_len = 128) | +3.1% |
> > >
> > > This overhead is small compared to the savings from sparse execution at long context. In other words, the controller cost is not what determines practicality here.
> > >
> > > ### Dynamic vs fixed: same average budget, same speed-up, better quality
> > >
> > > The key comparison is **model quality on dynamic sparse vs fixed sparse at the same average budget**.
> > >
> > > If both methods use the same average $(\kappa, \rho, q)$, then they operate at the **same average compute budget** and therefore roughly the same practical speed-up regime. The difference is that **SOL allocates that budget non-uniformly across tokens**, which is exactly why it gets better model quality.
> > >
> > > Across matched configurations:
> > >
> > > - **Llama-3.1-8B-Instruct:** dynamic wins in **88.9%** of comparisons (**504 / 567**)
> > > - **Llama-3.2-3B-Instruct:** dynamic wins in **95.5%** of comparisons (**421 / 441**)
> > >
> > > Representative matched examples are below (N = number of samples for that configuration):
> > >
> > > | $\kappa$ | $\rho$ | q-bits | Dynamic PPL | Fixed PPL | $\Delta$PPL | Speedup @8K | Speedup @32K | N |
> > > |:-:|:-:|:-:|:-:|:-:|:-:|:-:|:-:|:-:|
> > > | 0.3 | 0.5 | 6 | 9.27 | 9.88 | **+0.62** | 1.90x | 2.45x | 12 |
> > > | 0.3 | 1.0 | 6 | 7.77 | 8.37 | **+0.60** | 1.42x | 1.93x | 10 |
> > > | 0.4 | 0.6 | 6 | 8.18 | 8.54 | **+0.36** | 1.71x | 2.15x | 3 |
> > > | 0.5 | 0.6 | 8 | 7.85 | 8.08 | **+0.23** | 1.60x | 1.89x | 2 |
> > > | 0.5 | 1.0 | 6 | 7.49 | 7.74 | **+0.25** | 1.33x | 1.69x | 4 |
> > >
> > > Here, $\Delta$PPL = Fixed PPL $-$ Dynamic PPL, so **positive means dynamic is better**.
> > >
> > > This is the central practical point: **at the same realized budget, dynamic methods like SOL lead to a similar speed-up regime while giving better perplexity**.
> > >
> > > We hope this resolves the remaining concern. We present both sides of the claim:
> > > 1. **practical sparse operating points do yield speed-ups**, and
> > > 2. **dynamic policies are better than fixed policies at those same budgets**.
> > >
> > > We would be very grateful if the reviewer could reconsider their rating in light of this additional evidence.

---

### Official Review · Reviewer_4B27 · 2026-03-10

**Soundness:** 3
**Presentation:** 2
**Significance:** 3
**Originality:** 2
**Overall Recommendation:** 4
**Confidence:** 1

**Summary:**

This paper studies how to allocate inference-time computation more effectively during autoregressive generation. The main idea is that some tokens are easy to predict, while others are more sensitive to approximation. Furthermore, it becomes more complicated that approximation errors at a given step can be preserved in the KV cache and can influence later predictions. To address this, the paper introduces a controller that can select the compute budget for each token by adjusting attention sparsity, activation pruning, and quantization. The experiments are comprehensive, and the results show consistent gains over fixed allocation under same budget. Overall, the paper is well motivated, easy to follow, and empirically convincing.

**Compliance With Llm Reviewing Policy:**

Affirmed.

**Final Justification:**

Overall, the paper is interesting and has provided many insights. I would like to keep my current score due to my limited knowledge in this area.

**Key Questions For Authors:**

Please see weaknesses.

**Limitations:**

yes

**Strengths And Weaknesses:**

Strengths:
1. The studied problem is important.
2. The motivation is straightforward and sound, Figure 7 is insightful.
3. Extensive ablations are performed to understand the effects of action space, horizon length, budget control, and different scheduling strategies.

Weaknesses:
1. While the controller is lightweight and the paper argues that periodic KV refresh has limited overhead due to higher prefill throughput, the paper does not provide a thorough quantitative analysis of the additional system cost, such latency, or FLOPs overhead introduced by the controller and refresh steps.
2. The paper does not report the cost of training the controller, such as GPU hours.

---

> ### Author Rebuttal · Authors · 2026-03-30
>
> Thank you for your feedback, please find our responses below:
>
> > While the controller is lightweight and the paper argues that periodic KV refresh has limited overhead due to higher prefill throughput, the paper does not provide a thorough quantitative analysis of the additional system cost, such latency, or FLOPs overhead introduced by the controller and refresh steps.
>
> Thank you for raising this important question, we agree that a hardware-aware latency/FLOPs analysis would strengthen the paper and we will clarify this more explicitly. However, a few points are important in this context.
>
> - Periodic KV refresh **is not intrinsic to SOL**It is a tunable mechanism to bound KV-pollution with fixed-horizon episodes, and SOL remains effective across different refresh periods. This allows us to study SOL with respect to fixed baselines in a constrained setting, and present our results. Our findings actually show that SOL improves in performance relative to fixed baselines as the refresh period is increased (as the quality-efficiency trade-off matters more at longer decode tasks). Further, the controller is lightweight, and adds constant per-token overhead.
> - As the refresh period / decision horizon increases, SOL’s advantage over the fixed baseline becomes larger, not smaller. For the same evaluation setup: ( ΔPPL = policy PPL − fixed PPL. )
>   - **T = 4:** ΔPPL = -0.127 ± 0.748, Cohen’s d = -0.17, policy wins 58.7%
>   - **T = 16:** ΔPPL = -0.160 ± 0.696, Cohen’s d = -0.23, policy wins 70.4%
>   - **T = 64:** ΔPPL = -0.212 ± 0.259, Cohen’s d = -0.82, policy wins 84.7%
>
>
>
> - The extra forward pass is not equivalent to T additional autoregressive decode steps, its a short dense pass over already known tokens, a prefill/verification style operation. This is why chunked/hybrid prefills, speculative decoding + self-speculative decoding and recent KV-refresh methods can add verification or recomputation while still improving end-to-end latency. We view refresh as an optional cache-repair mechanism which has substantially higher throughput than sequential decode.
>
> More broadly, our goal is to establish the algorithmic value of dynamic per-token budget allocation across a unified action space. Some operating points we study, such as per-step mixed low-bit activation quantization combined with pruning and token sparsity do not have mature deployable kernels. Thus, hardware-specific latency numbers will conflate our proposed control framework. We will revise the paper to make this more explicit, and highlight end-to-end kernel-aware evaluation and reward formulations as important future work.
>
>
> > The paper does not report the cost of training the controller, such as GPU hours.
>
> The cost to train the controller is reasonable, compared to the deployment and inference costs of large language models. We isolate the training runs and provide the GPU hours required for training for H=16 below. We will also add these to our final manuscript.
>
> - Llama-3.1-1B: 4 hours on H100
> - Llama-3.2-3B: 7 hours on H100
> - Llama-3.1-8B-Instruct: 20 GPU hours on H100

---

> > ### Author Rebuttal · Reviewer_4B27 · 2026-04-01
> >
> > All of my concerns have been addressed. I would like to keep my current score due to my limited knowledge in this area.

---

> > > ### Author Response · Authors · 2026-04-06
> > >
> > > Thank you for your positive score!

---

### Official Review · Reviewer_aoV7 · 2026-03-10

**Soundness:** 3
**Presentation:** 3
**Significance:** 3
**Originality:** 4
**Overall Recommendation:** 5
**Confidence:** 3

**Summary:**

The paper introduces a method for dynamic allocation of inference-time compute in LLMs for each token generation. This method involves training a policy network to select, in each step, the amount of attention sparsity, MLP activation pruning, and activation quantization bit-width to use. The base LLM remains frozen. The authors evaluate the method on llama 3.1 and 3.2, with the model size ranging from 1B to 8B. They compare the method against static baselines and report improved perplexity-efficiency Pareto frontiers, and gains on downstream benchmarks.

**Compliance With Llm Reviewing Policy:**

Affirmed.

**Final Justification:**

The paper's idea of per-token compute allocation is novel and can be built upon in subsequent work. Taking the novelty and the rebuttal into consideration, I am raising my score to a 5. My chief remaining concern is that of real-world efficiency gains, but I understand it is perhaps beyond the scope of this paper. The paper would benefit from a more explicit discussion on limitations and future work.

**Key Questions For Authors:**

1. Net keep-rate is a simple average of the individual keep-rates for sparsity, pruning, and quantization. This assumes equal contribution to compute cost. Is this assumption justified?
2. How sensitive are results to the penalty weights and tolerance band?
3. Have you considered applying the method to MoE architectures? It would be helpful to understand how the improvements translate to these newer models.
4. Could you report results on more downstream benchmarks like IFEval and HumanEval?

**Limitations:**

A more explicit discussion of real-world deployment challenges would strengthen this section.

**Strengths And Weaknesses:**

**Strengths:**
1. The problem statement is well-motivated.
2. The methodology is well-thought-out, with a thorough discussion on and remediation of delayed effects of efficiency actions (KV Pollution).

**Weaknesses:**
1. The proxy for efficiency in the paper is “net keep-rate”. It does not directly correspond to latency, throughput, or memory usage. It is unclear what real-world gains would be realized by this approach given:
    - Memory allocation appears unchanged (since it must account for the worst case scenario, which is when we decide to take none of the efficiency actions). This is a disadvantage compared to some of the static methods that would guarantee reduced memory usage.
    - We must now run an extra forward pass once every T=16 steps.
    - Per-token quantization perhaps complicates kernel design.
2. The method introduces three target budget parameters, three penalty weights, and a tolerance band parameter. However, it is unclear how to choose these parameters, and how robust is the performance to these choices.

---

> ### Author Rebuttal · Authors · 2026-03-30
>
> We would like to thank the reviewer for their feedback. Please find our responses below.
>
> > The proxy...“net keep-rate”....  realized by this
>
> In this paper, we propose a framework to learn from efficiency feedback while preserving the model quality. To demonstrate this effectively, we present our results with ‘net keep-rate’, which allows us to visually interpolate between quantization, pruning and token sparsity. Depending on the hardware-software stack, the trade-off in these efficiency methods can change (with quest page-based sparsity, token sparsity can be greatly accelerated, but not so for fine-grained sparsity, in which case, quantization or structured pruning may be a better option). To focus on the core idea, we treat these details of “how to construct efficiency reward” as out of scope, and instead compare with existing methods directly, in terms of ‘observed keep rates’.
>
> > memory allocation appears unchanged
>
> Much like FP8 inference, we can always specify the action space with a more constrained set of options, i.e., instead of having quantization options as [4, 8, 16], one can constrain to [4,8] if desired. Thus, our memory allocation logic is exactly equivalent to the corresponding static method, where we can construct the action space to have the same guarantees.
>
> > run an extra forward pass every T=16 steps
>
> This is done largely for presentation (to bound perplexity measurement), as shown in our response to Reviewer ms4V, the performance of SOL actually gets better as the T increases, with T=4 only giving 58% win-rate, but going up to 84.7% win-rate with T=64. Much like speculative decoding, ‘fast prefill’ has significantly higher throughput than decode, so we do not expect this to be a major bottleneck even if implemented.
>
> > per-token quantization and kernel design
>
> We completely agree, our framework demonstrates three different efficiency methods, page-based sparsity for e.g., is compatible with current deployment frameworks. **Dynamic decode-time quantization does not yet have a kernel, as dynamic per-step budget allocation has not been previously studied sufficiently.** Our goal in this paper was to provide the highest coverage of efficiency options and demonstrate clear gains of dynamic budget allocation, which we hope inspire relevant kernel design which is out of scope for this paper.
>
> > The method…  and how robust is the performance to these choices
>
> The target budget parameters are randomly sampled during training to get a robust policy. The tolerance band depends on the deployment setting i.e., how strongly must the performance SLO be met (we use a very tight tolerance, as seen in Figure 4). Most importantly, we agree that the three penalty weights are indeed hyper-parameters, in our experiments, we calibrated them so that the contribution of budget-violation penalties was roughly 1:1 with the model-quality loss. We do not see a reason to expect a single universally optimal choice of these weights as we scalarize an inherently multi objective tradeoff in quality vs budget adherence. We however empirically demonstrate this 1:1 weighting consistently outperforms existing fixed baselines.
>
> > Net keep-rate is a simple ... on justified?
>
> We believe this assumption is justified for the purposes of presentation. I.e., the policy can still adhere to arbitrary budget requests that are not the ‘average’ keep-rates. This is why we present pareto plots with thousands of perplexity measurements on a 2D plot. This is further supported by our table in response to reviewer ms4V, where we bin by per-efficiency-axis instead of just the average and show that we beat fixed methods with very high statistical significance.
>
> > Sensitivity to penalty weights a... MoE architectures
>
> All our policies were trained on different action spaces, horizons and model scales, with no requirement to calibrate the penalty weights and tolerance, when set reasonably (1:1 with model quality), the policy trains well. However, there definitely is scope for hyper-parameter optimization. We do not have the resources to train on larger MoE models, we hope the reviewer understands. However, intuitively speaking experts should not change the dynamic budget allocation tradeoff with model quality.
>
> > Could you...HumanEval?
>
>  Llama-3.1-8B-Instruct
> ### HumanEval-Instruct (pass@1)
>
> | Random Config (κ / ρ / bits) | Policy | Fixed | Dense |
> |---|---|---|---|
> | Dense baseline | — | — | 70.12 |
> | 0.95 / 0.90 / 13 | 70.52 |  68.42 | — |
> | 0.85 / 0.90 / 11 | 67.07 |  64.63 | — |
> | 0.75 / 0.90 / 8  | 66.46 |  62.20 | — |
> | 0.65 / 0.90 / 10 | 61.59 |  55.49 | — |
>
> ### IFEval:
>
> | Random Config (κ / ρ / bits) | Policy | Fixed | Dense |
> |---|---:|---:|---:|
> | Dense baseline | — | — | 80.30 |
> | 0.65 / 0.95 / 10 | 80.30 | 78.34 | — |
> | 0.75 / 0.90 / 8 | 76.38 | 70.51 | — |
> | 0.85 / 0.90 / 11 | 80.30 | 72.47 | — |

---

> > ### Author Rebuttal · Reviewer_aoV7 · 2026-04-02
> >
> > Thank you for the added explanation and as well as running additional experiments.
> >
> > > Much like FP8 inference, we can always specify the action space with a more constrained set of options, i.e., instead of having quantization options as [4, 8, 16], one can constrain to [4,8] if desired. Thus, our memory allocation logic is exactly equivalent to the corresponding static method, where we can construct the action space to have the same guarantees.
> >
> > While that is true, in the paper the experiments compare `[4, 8, 16]` against `8` (ZeroQuant-8bit) and not `[4, 8]` against `8`.

---

> > > ### Author Response · Authors · 2026-04-02
> > >
> > > Thank you for the follow up question.
> > >
> > > Yes, we compare [4, 8, 16] with 8-bit ZeroQuant, this is because we are trying to present that 'you can use a wide range of bit-precision, and benefit from improvement in model quality'. However, fundamentally, there is no limitation in the quantization search space. To provide further evidence of this, we take the entire evaluation search space reported in Figure 5 (1521, 442, 567) points for the 1B, 3B and 8B models and 'filter out' any evaluations where P(16 bit) > 0.01 for the entire evaluation run. This removes every operating point where the policy had any meaningful reliance on 16-bit precision.
> > >
> > > The policy already rarely uses 16-bit. Across all evaluation points, the average marginal probability of selecting 16-bit quantization is just 4.4% (1B), 4.2% (3B), and 10.7% (8B). The majority of evaluation points already satisfy P(16-bit) < 1%: 888/1521 (58%) for 1B, 292/441 (66%) for 3B, and 284/567 (50%) for 8B. The dominant quantization choices are 6-8 bit across all models.
> > >
> > > ## Results (P(16-bit) < 1% filter, best SOL operating points)
> > >
> > > | Model | Best SOL PPL | ZQ-8bit PPL | Δ       | SOL eff. quant bits | SOL token keep | SOL head keep | SOL net resource (x) | ZQ-8bit net resource |
> > > |-------|-------------|-------------|---------|---------------------|----------------|---------------|----------------------|----------------------|
> > > | 1B    | 9.833       | 9.796       | +0.4%   | 8.65                | 0.792          | 0.989         | 0.774                | 0.833                |
> > > | 3B    | 7.776       | 7.780       | **−0.1%** | 8.87              | 0.964          | 0.918         | 0.812                | 0.833                |
> > > | 8B    | 7.120       | 7.139       | **−0.3%** | 8.04              | 0.920          | 0.959         | 0.794                | 0.833                |
> > >
> > > On both 3B and 8B models, SOL achieves lower perplexity than ZeroQuant-8bit while never using 16-bit quantization, and also simultaneously evicting token and pruning attention heads, resources that ZeroQuant keeps at 100%! On the 1B model, SOL comes within 0.4% of ZeroQuant-8bit PPL at 7.1% lower total usage!
> > >
> > > On ZeroQuant-bit the advantages are even more pronounced, with P(16-bit) < 1%, SOLs best operating points achieve 12% lower PPL (1B), 7.5% lower PPL (3B) and 24.9% lower PPL on 8B, with 98% win-rate on ZeroQuant-5bit for the 8B model. This win-rate does not even consider the fact that SOL simultaneously optimizes token-sparsity and pruning rates.
> > >
> > > ## vs ZeroQuant-5bit (P(16-bit) < 1% filter)
> > >
> > > | Model | Best SOL PPL | ZQ-5bit PPL | Δ PPL    | % points beating ZQ-5bit |
> > > |-------|-------------|-------------|----------|--------------------------|
> > > | 1B    | 9.810       | 11.143      | −12.0%   | 49.3% (438/888)          |
> > > | 3B    | 7.752       | 8.383       | −7.5%    | 36.6% (107/292)          |
> > > | 8B    | 7.120       | 9.477       | −24.9%   | 97.9% (278/284)          |
> > >
> > > The key insight is that SOL can indeed guarantee reduction in total resource utilization by either training with a limited design space (e.g., remove 16-bit) or simply sampling actions in the desired sub-set of the keep rate. Even with restricted 8-bit effective precision, non-uniform allocation is able to find better quality-resource trade-offs than any single static bit-width. SOL literally beats ZQ-8bit PPL with q16 < 1%, with even larger gains over ZQ-5bit.
> > >
> > > We hope this clarifies that SOL is naturally capable of sampling actions which are not 'full precision', thereby guaranteeing adherence to desired budgets. Please let us know if there are any further doubts we can provide clarification for.

---

### Official Review · Reviewer_ms4V · 2026-03-13

**Soundness:** 2
**Presentation:** 2
**Significance:** 3
**Originality:** 3
**Overall Recommendation:** 4
**Confidence:** 2

**Summary:**

The paper addresses the problem of assigning a per-token budget along the axes of token-level attention sparsity, activation pruning, and quantization, using an RL-based formulation.

**Compliance With Llm Reviewing Policy:**

Affirmed.

**Final Justification:**

I thank the authors for the additional clarifications. In light of the experiments and the clarifications provided, I am increasing my score to 4. That said, I am reducing my confidence, as I still have significant uncertainty about parts of the submission and about whether I have the full background needed to evaluate it in depth. I also continue to find the paper difficult to follow, and I believe it would benefit from a substantial rewrite and clearer, more informative figures.

**Key Questions For Authors:**

**Questions:**

1.	Could the authors explain what the net keep rate refers to? I do not recall seeing it clearly defined in the paper.

2.	Could the authors explain how the graphs should be read? The captions do not seem sufficiently informative, and I found it difficult to interpret what each figure is showing.

3.	Could the authors report standard deviations for the main results?

4.	Could the authors clarify which figure supports the claim that SOL achieves 7.3% higher accuracy than uniform budget allocation strategies?

5.	If I understand correctly, the method first generates a sequence of T tokens, each with its own quantization, attention sparsity, and related settings, and then regenerates these T tokens without those modifications. If that understanding is correct, could the authors clarify how this procedure leads to efficiency gains?

6.	Could the authors clarify what is meant by the hidden state of the LLM? Also, when attention sparsity, quantization, and related mechanisms are defined, are they applied uniformly across all transformer layers? If so, would a per-layer formulation be more appropriate?

**Note:** I am currently leaning toward a lower score, but I would be open to revising my assessment depending on the rebuttal.

**Limitations:**

Yes.

**Strengths And Weaknesses:**

**Strengths:**

1. The paper focuses on an important problem.

**Weaknesses:**

1.	The paper was somewhat hard to read, as several details seemed to be missing.

2.	The graphs in the paper were generally hard to interpret.

---

> ### Author Rebuttal · Authors · 2026-03-30
>
> Thank you for your thorough review! Please find our responses below
>
> > Could the authors explain what the net keep rate refers to?
>
> We apologize for not defining this clearly, and will revise the paper. By “Net Keep Rate,” we mean the realized average keep-rate over an episode. For example, if token keep-rates over 4 steps are (10%, 50%, 50%, 90%), the net keep-rate is 50%. For quantization, averaging bit-widths (4, 8, 8, 4) gives 6 bits, or 37.5% relative to 16-bit precision. When multiple efficiency methods are combined, we report the overall Net Keep Rate as the average of the realized keep-rates of each method.
>
> > Explain how the graphs should be read
>
> Figures 2 and 3 should be read as: given a target budget (Net Keep Rate), what perplexity is achieved, and what realized Net Keep Rate does the policy attain? In both figures, the policy finds a better quality-efficiency tradeoff than static baselines. Figure 2 shows this remains true with a larger joint action space, while Figure 3 shows it remains true over longer episodes.
> Figure 4 shows how closely the policy meets the requested budget. Figure 5 compares our perplexity results to prior token-sparsity, pruning, and quantization methods. Figure 6 evaluates the policy on downstream tasks: log-likelihood, completion, and reasoning decode settings.
>
> > Could the authors report std-dev
>
> To check whether the policy’s improvement is actually meaningful, we also ran a statistical analysis over all evaluated configurations. Each row in the sweep corresponds to a target (keep, prune, quantization) triplet, and both the learned policy and fixed baseline are evaluated at the same target. We report mean ± standard deviation in perplexity for both, along with the paired difference Δ = policy PPL − fixed PPL. (Repeated model entries dropped)
>
> | Config | Policy PPL | Fixed PPL | Δ (policy − fixed) | p-value | Cohen's d | % policy wins |
> |:-|:-|:-|:-|:-|:-|:-|
> | SOL-J-FL-T4 | 9.53 ± 1.87 | 9.66 ± 2.20 | -0.127 ± 0.748 | 3.2e-05 | -0.17 | 58.7% |
> | SOL-J-FL-T16 | 11.26 ± 1.29 | 11.42 ± 1.60 | -0.160 ± 0.696 | 3.0e-08 | -0.23 | 70.4% |
> | SOL-J-FL-T64 | 9.12 ± 0.64 | 9.34 ± 0.79 | -0.212 ± 0.259 | <1e-10 | -0.82 | 84.7% |
> | SOL-J-2L-T16 | 11.51 ± 0.60 | 11.89 ± 0.72 | -0.383 ± 0.241 | <1e-10 | -1.59 | 95.3% |
> | SOL-J-3L-T16 | 10.63 ± 0.77 | 10.94 ± 1.08 | -0.309 ± 0.346 | <1e-10 | -0.90 | 84.4% |
> | Llama-3.2-3B | 9.14 ± 1.19 | 9.46 ± 1.28 | -0.327 ± 0.243 | <1e-10 | -1.34 | 95.5% |
> | Llama-3.1-8B | 7.93 ± 0.62 | 8.20 ± 0.77 | -0.270 ± 0.240 | <1e-10 | -1.12 | 88.9% |
>
> Across all seven configurations reported, the learned policy achieves statistically significantly lower perplexity than the fixed baseline (one-sided paired t-test, p < 10^-4 in every case). The strength of this advantage generally increases with decision horizon: Cohen’s d grows from -0.17 at T = 4 to -0.82 at T = 64, and reaches -1.59 for the 2-layer variant. The policy also achieves lower perplexity in 59% to 95% of individual configurations, suggesting that the improvement is broad rather than driven by a small number of outliers. When we further group results by net keep-rate (within a ±2.5% tolerance), the policy advantage remains significant in most bins, and in all bins for the T = 64 setting.
>
> > 7.3% improvement
>
> This comes from: (47.561 - 44.313)/ 44.313, Figure 6 (Middle)
>
>
> > If I understand correctly, …  procedure leads to efficiency gains?
>
> If we understand correctly, the reviewer is referring to the periodic KV-cache refresh. This is not required by our method (our method gets stronger as the horizon grows, as shown in the table above), and we only use it to bound the episode for presentation. Over very long decode horizons, very low keep-rates can gradually corrupt the KV-cache and hurt performance; this is a general issue with sparse/quantized inference, not something specific to our policy. Since prefill is much faster than decode, this refresh is optional and relatively cheap.
>
> > ... hidden state of the LLM? ... per-layer formulation ...?
>
> We mean the last-layer representation of the current token, i.e., the output of the final transformer block before the LM head produces logits.
> Efficiency methods are applied uniformly across layers. We agree that a per-layer formulation is likely more appropriate, and there is prior work on this, but it is outside the scope of this paper. Our focus here is adaptation across decode steps, not layers.
>
> > Note: I am ... open to revising my assessment depending on the rebuttal.
>
> We sincerely thank the reviewer for their time and feedback. SOL takes a first step toward RL environments for self-optimizing language model decoding, and our results show this is a strong direction: across all seven configurations in the table above, the learned policy significantly outperforms the fixed baseline, with lower perplexity in 59%–95% of cases.

---

> > ### Author Rebuttal · Reviewer_ms4V · 2026-04-01
> >
> > Thank you for the rebuttal. I still have a few follow-up questions:
> >
> > **Q1.** Why is net keep-rate the main efficiency metric in the paper? Since it is only a proxy for compute, would latency, throughput, memory, or storage be more informative for the practical claims being made?
> >
> > **Q2.** Could the authors clarify why the method does not fundamentally rely on periodic KV-cache refresh? If refresh is reduced or removed, how does performance change? Relatedly, how should one think about the inference-time distribution shift introduced by low-bit quantization and other aggressive efficiency actions, given that these behaviors are not part of standard pretraining?
> >
> > **Q3.** Please make it explicit that the reported 7.3% gain is a relative improvement, not an absolute one.
> >
> > **Q4.** For the discussion of using the LLM hidden state and the possible per-layer formulation, could the authors more clearly explain what prior work exists in this direction and why that design space is outside the scope of the present paper?
> >
> > **Comments.** The paper would benefit from substantial rewriting for clarity. It may also help to revise the title so that it more accurately reflects the actual contribution.

---

> > > ### Author Response · Authors · 2026-04-01
> > >
> > > Thank you for further feedback, we will try to make the writing clearer.
> > >
> > > Q1: Why is net keep-rate the main efficiency metric in the paper.
> > >
> > > We are proposing a framework that enables a policy to trade-off between several efficiency methods. For e.g., token-sparsity can have multiple implementations (page sparsity, block sparsity, etc.), similarly quantization can be implemented either with quantized KV-Cache states or just activation quantization, depending on the goal (latency, throughput, memory). We believe the most unbiased (and simple) metric to report is simply 'what fraction of full-capacity is preserved', which is the net keep-rate. Also note that since our proposal is on multiple efficency methods, there is no single implementation that we are aware of, that can accelerate all three well (especially fine-grained weight pruning). We believe presenting a reasonably wide suite of efficiency methods on our method was more important than demonstrating hardware speed-up over every method (as we are trying to focus on the algorithmic value of being able to learn nuanced contextual and dynamic decode-time efficiency-quality trade-offs).
> > > However, in response to reviewer XsE9, we present further performance results on a real quest page-based sparsity implementation. We present a part of this result below as well (We profile **Llama-3.1-8B-Instruct** on **H200 140GB, FP16**):
> > >
> > > | Keep Fraction ($\kappa$) | Speedup @1K | Speedup @8K | Speedup @32K |
> > > |:-:|:-:|:-:|:-:|
> > > | 0.2 | 1.10x | 1.39x | 1.85x |
> > > | 0.3 | 1.07x | 1.32x | 1.66x |
> > > | 0.4 | 1.05x | 1.25x | 1.50x |
> > > | 0.5 | 1.04x | 1.19x | 1.37x |
> > >
> > > We could indeed report latency on each axis, but we would then have to analytically compare the trade-off in these three methods (quantization, weight pruning and activation-based token sparsity), which would look very similar to our net keep-rate in nature.
> > >
> > > Q2: Periodic KV-Cache refresh
> > >
> > > It is for mostly two reasons; (a) to train on bounded episode lengths and (b) keep the policy itself constant overhead. In reality, we can construct arbitrarily long episodes, and since the policy itself is a small autoregressive transformer, it is compatible. In Figure 3, the pareto front discovered with a longer episodes are actually better, as also shown in our rebuttal Table. This is intuitive, as the longer we decode with say, quantization, the more likely we are to encounter opportunities to use 'lower compute' on some tokens whereas 'higher compute' on the others. The win-rate for the policy over fixed actually increases. If the reviewer requests, we can launch RL for another policy with unbounded (no refresh), but note that nothing changes significantly from normal static decode, we already present results with 64 step long episodes.
> > >
> > > Further, the inference-time distribution shift introduced by low-bit quantization should be the same for our policy, as it is for fixed strategies. I.e., if the user requests an aggressive target sparsity of 30%, then the optimal action for policy could indeed be to keep its sparsity fixed to 30% at every step. An efficiency target that is doomed to fail for fixed methods is very likely going to fail for the policy.
> > >
> > > Q3: Plase make it explicit that 7.3 % gain is relative
> > >
> > > Yes, we completely agree and are sorry that it came across incorrectly and can be misleading. We will change '7.3%' to '7.3% relative improvement'.
> > >
> > > Q4: Possible per-layer formulations
> > >
> > > [DejaVu, ShadowLLM] propose using hidden states to predict neuron saliency, similarly, [DSA] uses 'token indexers' to non-unfiromly select token budgets for each layer, with [SparCIM] trying to accelerate heterogeneous bit-sparsity. For these works, they require an 'efficiency target' that will dicate a threshold (for e.g., DSA can use a fixed token budget, or ShadowLLM can decide a threshold to hit 50% pruning rate), but essentially, this decision will be 'distributed' across all layers non-unfiromly. Thus, we believe that predictors that decide 'how much' to prune at each step are currently a bit orthogonal to predictions that decide 'which layers' should be preserved more. Technically, it is 100% true that SOL could be trained to start predicting per-head token sparsity rate, and that may give a much better result, but we do not know what the associated cost of a predictor with such a large output space will be, and whether it is feasible from a run-time practically perspective.
> > >
> > > We hope this helps clarify our design choices!
> > >
> > > [DejaVu] Liu, Z.,. Deja Vu: Contextual Sparsity for Efficient LLMs at Inference Time. https://arxiv.org/abs/2310.17157
> > >
> > > [ShadowLLM] Akhauri, Y., .... ShadowLLM: Predictor-based Contextual Sparsity for Large Language Models. https://arxiv.org/abs/2406.16635
> > >
> > > [DSA] DeepSeek-AI, .... (2025). DeepSeek-V3.2: Pushing the Frontier of Open Large Language Models. https://arxiv.org/abs/2512.02556
> > >
> > > [SparCIM] Xu, X., Song, Y....  & Liu, B. (2025). SparCIM: ... . 2025 (ICCAD), 1–9. https://doi.org/10.1109/ICCAD66269.2025.11240806

---

### Decision · Program_Chairs · 2026-04-30

**Decision:**

Accept (regular)

**Comment:**

The paper introduces a method for dynamic per-token compute allocation in language model decoding, where a small policy network selects, at each generation step, the levels of attention sparsity, MLP activation pruning, and activation quantization applied to a frozen base model.

There was consensus among reviewers that this is an interesting and important contribution, with all four recommending acceptance. In the revised version, the authors are encouraged to carefully address all comments raised by the reviewers by incorporating the additional arguments and results they provided during the rebuttal. The most consistent concern is that the main efficiency measure, the net keep rate, is only a proxy for practically relevant quantities such as latency, throughput, or memory usage. The authors should incorporate the end-to-end speed-up measurements and overhead numbers from the rebuttal, along with the downstream results on HumanEval and IFEval. On the presentation side, the paper would benefit from an explicit definition of the net keep rate, and more informative and comprehensive figure captions. The authors should also consider revising the title, since "self-optimizing" does not convey that the goal is inference efficiency rather than task performance.